EMBO
*reports*

# Report

# O-glycan initiation directs distinct biological pathways and controls epithelial differentiation

Ieva Bagdonaite[1], Emil MH Pallesen[1], Zilu Ye[1], Sergey Y Vakhrushev[1], Irina N Marinova[1], Mathias I Nielsen[1], Signe H Kramer[2], Stine F Pedersen[2], Hiren J Joshi[1], Eric P Bennett[1,3], Sally Dabelsteen[3] & Hans H Wandall[1,*] iD

## Abstract

Post-translational modifications (PTMs) greatly expand the function and potential for regulation of protein activity, and O-glycosylation is among the most abundant and diverse PTMs. Initiation of O-GalNAc glycosylation is regulated by 20 distinct GalNAc-transferases (GalNAc-Ts), and deficiencies in individual GalNAc-Ts are associated with human disease, causing subtle but distinct phenotypes in model organisms. Here, we generate a set of isogenic keratinocyte cell lines lacking either of the three dominant and differentially expressed GalNAc-Ts. Through the ability of keratinocytes to form epithelia, we investigate the phenotypic consequences of the loss of individual GalNAc-Ts. Moreover, we probe the cellular responses through global transcriptomic, differential glycoproteomic, and differential phosphoproteomic analyses. We demonstrate that loss of individual GalNAc-T isoforms causes distinct epithelial phenotypes through their effect on specific biological pathways; GalNAc-T1 targets are associated with components of the endomembrane system, GalNAc-T2 targets with cell–ECM adhesion, and GalNAc-T3 targets with epithelial differentiation. Thus, GalNAc-T isoforms serve specific roles during human epithelial tissue formation.

**Keywords** 3D skin; differential glycoproteomics; polyomics; polypeptide GalNAc-transferase; tissue development

**Subject Categories** Post-translational Modifications & Proteolysis; Proteomics; Skin

## Introduction

Glycosylation is an abundant post-translational modification of both intracellular and extracellular proteins [1]. The majority of glycans are classified as N-linked chains, where the carbohydrate moiety is attached to asparagine residues, or O-linked chains, most commonly linked to a serine or threonine. N-linked glycosylation is initiated by the oligosaccharyltransferase complex with only two paralogs of the catalytic subunit, whereas O-glycan initiation is more complex. There are several types of O-linked glycosylation, but among the most diverse is the mucin or GalNAc type (hereafter referred to as O-glycosylation). O-glycosylation is initiated by 20 evolutionarily conserved polypeptide GalNAc-transferases (GalNAc-Ts), which add GalNAc residues to serine and threonine amino acids (Fig 1A). Each of the GalNAc-Ts is differentially expressed in various tissues and has both distinct and overlapping peptide substrate specificities [2–12]. Thus, the repertoire of GalNAc-Ts expressed in a given cell determines the subset and O-glycosite pattern of glycosylated proteins [13]. Substantial efforts have been made to characterize and predict the substrate specificities of GalNAc-Ts *in vitro,* but understanding of the *in vivo* specificities of the individual GalNAc-Ts or their biological functions is limited [13–15]. This lack of insight prevents an understanding of how site-specific O-linked glycosylation affects diseases, such as metabolic disorders, cardiovascular disease, and various malignancies, that have been associated with GalNAc-Ts through genome-wide association studies and other linkage studies [16–26]. Therefore, it is imperative that we establish how O-glycosylation at specific sites in proteins affects protein function.

We and others recently developed new strategies for identifying specific sites on proteins that undergo O-glycosylation in different cell types and tissues [27–31]. Characterization of the O-glycoproteomic landscape in isolated human cells and multiple human cell lines suggests that more than 80% of all proteins that traffic through the secretory pathway are O-glycoproteins [28,30]. Probing the non-redundant contributions of individual GalNAc-Ts in cells lacking specific enzymes [32–34] has revealed broad substrate specificities for some of the individual isoforms and very restricted specificities for others [33–35]. Assessing all of the mapped O-glycosylation sites to identify associations between O-glycosites and protein annotations, we recently found that O-glycans are over-represented close to tandem repeat regions, protease cleavage sites, within propeptides, and on a select group of protein domains [28,30,36]. Although such general associations between the location of O-glycans and

1   Copenhagen Center for Glycomics, Institute of Cellular and Molecular Medicine, University of Copenhagen, Copenhagen, Denmark
2   Cell Biology and Physiology, Department of Science, University of Copenhagen, Copenhagen, Denmark
3   School of Dentistry, University of Copenhagen, Copenhagen, Denmark
    *Corresponding author. Tel: +45 27210936; E-mail: hhw@sund.ku.dk

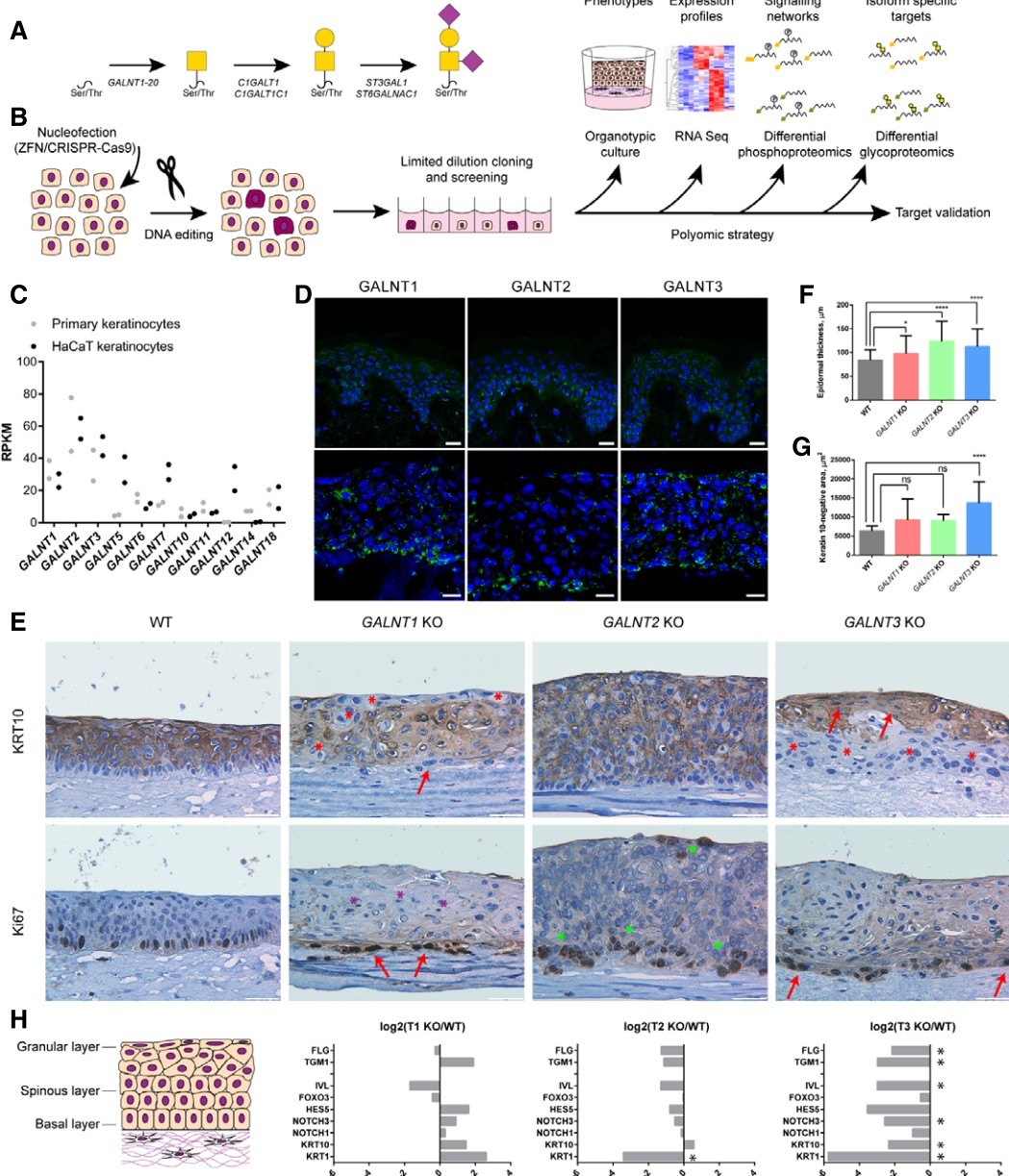

**Figure 1. Phenotypic characterization.**

A    O-GalNAc-type O-glycosylation pathway. Biosynthesis of core 1-type structures is shown.

B    Strategy for generation and characterization of *GALNT* isoform knock outs in HaCaT keratinocytes.

C    Expression of *GALNT* isoforms in primary keratinocytes and HaCaT cell line. The scatter plot depicts individual RPKM values of 2 biological replicates.

D    Expression of GalNAc-T1, GalNAc-T2, and GalNAc-T3 in human skin (upper panel) and HaCaT keratinocyte organoids (lower panel). Frozen human skin or HaCaT keratinocyte organotypic skin models were stained using antibodies for the GalNAc-T isoforms. Scale bar—25 μm.

E    Phenotypic characterization of organotypic models made with HaCaT WT or *GALNT1/2/3* KO keratinocytes. IHC of tissue sections stained for differentiation marker keratin 10 (upper panel) or proliferation marker Ki67 (lower panel). Scale bar—50 μm. Red arrows—flattened cells; red asterisks—K10-negative region in suprabasal/granular layers; purple asterisks—pyknotic nuclei; and green asterisks–increase in Ki67-positive cells.

F    Quantification of epidermal thickness of skin organotypic models. Epidermal thickness was measured in 5 distinct images (4 positions/image) of 4 clones of *GALNT* isoform KO or WT (4 different tissues) and is presented as averages +SD. Due to high ZFN KO phenotypic inter-clonal variation and to exclude off target effects, 5 clones of ZFN and 3 clones of CRISPR KO (each targeted by a different gRNA) were used for *GALNT3* KO. ANOVA followed by Dunnet's multiple comparison test was used to compare mean epidermal thicknesses of different KOs to WT. *$P \leq 0.05$; ****$P \leq 0.0001$.

G    Quantification of keratin 10-negative area in skin organotypic tissue sections. Keratin 10-negative area was measured in 5 distinct images of 4 clones of *GALNT* isoform KO or WT (4 different tissues) and is presented as averages +SD. Due to high ZFN KO phenotypic inter-clonal variation and to exclude off target effects, 5 clones of ZFN and 3 clones of CRISPR KO (each targeted by a different gRNA) were used for *GALNT3* KO. ANOVA followed by Dunnet's multiple comparison test was used to compare mean areas of different KOs to WT. ****$P \leq 0.0001$.

H    RNAseq expression data of select differentiation markers, expressed in spinous and granular layers of the epidermis (cartoon). Expression data are presented as log$_2$(average KO/average WT). Significantly changed transcripts, identified in R analysis, are marked with asterisks.

protein functions may direct future investigations, the strategy does not define the function of site-specific glycosylation. Further progress in discovering and defining novel functions of site-specific glycosylation events requires direct quantitative analysis of potential biological responses induced by the loss of distinct GalNAc-T isoforms, and such biological responses are not easily observed in single cell culture systems. We previously used an organotypic tissue model equipped with genetically engineered cells to decipher the function of elongated O-glycans [29]. In the present study, we took advantage of this complex organotypic model of human epithelia and combined it with quantitative O-glycoproteomics and phosphoproteomics to search for biological functions of site-specific O-glycosylation (Fig 1B). With this combinatorial strategy, we demonstrate that loss of individual GalNAc-T isoforms has distinct phenotypic consequences through their effect on selective biological pathways.

## Results and Discussion

### Site-specific glycosylation influences epithelial homeostasis and differentiation of human skin cells

To probe the non-redundant functions of individual GalNAc-Ts, we developed an isogenic cell model in spontaneously immortalized human epidermal keratinocytes [37]. HaCaT cells undergo normal differentiation in organotypic cultures to form a stratified squamous epithelium and express O-glycans in a differentiation-dependent manner similar to human skin [29,37]. The model provides a strong tool to analyze the phenotypic consequences of losing individual GalNAc-Ts and the effects on defined biological functions, including proliferation, differentiation, cell–cell interactions, cell–matrix interactions, and signaling. First, we characterized the GalNAc-T repertoire in the HaCaT cell line by RNA sequencing and immunocytochemistry, demonstrating that the expression profiles of *GALNT* genes are comparable to human skin (Fig 1C and D). Immunocytochemistry showed the localization of GalNAc-T1, GalNAc-T2, and GalNAc-T3; human skin and HaCaT 3D models expressed GalNAc-Ts in a similar expression pattern, with GalNAc-T2 primarily expressed in basal cells and broader expression of GalNAc-T1 and GalNAc-T3 in all epithelial layers (Fig 1D). To investigate the importance of GalNAc-T1, GalNAc-T2, and GalNAc-T3 in the differentiation of human skin, we used ZFN nucleases and CRISPR/Cas9 to generate isogenic HaCaT cell lines with loss of GalNAc-T1 (ΔT1), GalNAc-T2 (ΔT2), or GalNAc-T3 (ΔT3) (Fig 1B). Successful targeting of individual single cell clones was identified by detecting indels in amplicon analysis and validated by Sanger sequencing (Appendix Table S1). In addition, the elimination of GalNAc-T1, GalNAc-T2, and GalNAc-T3 was confirmed by immunocytochemistry using mAbs for the individual enzymes (Fig EV1). RNAseq verified the reduction of the targeted GalNAc-Ts in relevant knockout (KO) cells with a limited influence on other GalNAc-Ts, except for a prominent increase in the expression of *GALNT5*, especially in *GALNT3* KO cells (Dataset EV1, Appendix Fig S2). In addition, we found no overall change in ST, T, STn, or Tn expression (Fig EV1).

We used the set of engineered keratinocytes to form human tissue in a 3D organotypic skin model, enabling us to examine how distinct isoforms of GalNAc-Ts affect epithelial formation.

Organotypic cultures generated from wild-type (WT) HaCaT keratinocytes had multiple layers of differentiating cells, with proliferation restricted to the single basal layer of K10-negative cells, though with low levels of terminal keratinization as described previously (Fig 1E). Lack of GalNAc-T1 caused a prominent phenotype with a distinct loss of normal architecture and changes in differentiation and stratification. ΔT1 tissue exhibited basal cells with flattened and spindle cell morphology (Fig 1E, red arrows). In addition, the stratum spinosum and granulosum were changed, with increased eosinophilic appearance, flattened cells, and to some extent pyknotic nuclei (Fig 1E, purple asterisks). Proliferating cells visualized by Ki67 staining were confined to the basal layer. Furthermore, several non-differentiated cytokeratin 10 (K10)-negative cell regions were found throughout the tissue, signifying delayed differentiation in some areas, whereas other areas appeared to exhibit increased differentiation (Fig 1E, red asterisks). In contrast, ΔT2 tissue presented with dramatically increased growth and height of the organotypic cultures (Fig 1E and F). An increased number of proliferating cells were observed, confined mostly to the basal layer (Fig 1E, green asterisks). No major change was seen in the differentiation pattern. Finally, the most significant changes were observed with elimination of GalNAc-T3. ΔT3 tissues had a significantly increased number of non-differentiating basal cells lacking K10 expression (Fig 1E, red asterisks; Fig 1G). In addition, the overall tissue architecture was changed, though the proliferative Ki67-positive cells were still confined to a single cell layer (Fig 1E). In addition, an abrupt change was observed between K10-negative and K10-positive cells, with the latter immediately exhibiting flattened nuclei (Fig 1E, red arrows).

Next, we performed RNA sequencing to compare the expression profiles of conventionally grown WT HaCaT cells and HaCaT cells without GalNAc-T1, GalNAc-T2, and GalNAc-T3 (Dataset EV1). Analysis of the differential transcriptomes using the Benjamini–Hochberg procedure at an FDR-adjusted $P$ value of 0.05 identified rather limited sets of targets that consistently changed in all three clones of each *GALNT* isoform KO compared to WT (Dataset EV1). In addition, we looked at a distinct gene set involved in keratinocyte differentiation, despite lacking significance in some of the isoform KOs (Fig 1H). The expression of markers defining different stages of skin differentiation confirmed that ΔT1 caused a slightly mixed phenotype with a mild increase in the expression of most differentiation markers (Fig 1H). In contrast, minor changes were observed in ΔT2 cells, with a slight decrease in differentiation markers consistent with a limited change in the differentiation pattern of basal cells. ΔT3 induced the largest transcriptomic changes, with significant downregulation of keratins (KRT1, KRT10, KRT13), transglutaminase 1 (TGM1), filaggrin (FLG), and involucrin (IVL), which are all associated with decreased differentiation consistent with the observed phenotype. In addition, we observed an effect on the expression of NOTCH receptors and the Notch signaling responsive gene HES5 (Fig 1H).

### Differential phosphoproteomic analysis reveals that individual GalNAc-Ts control distinct biological pathways

To appreciate the changes in signaling networks associated with knocking out individual *GALNT* isoforms, we performed differential proteomic and phosphoproteomic analyses of TMT 10-plex-labeled,

TiO$_2$-enriched ΔT1, ΔT2, and ΔT3 tryptic digests (2 clones of each) compared to WT by tandem mass spectrometry (MS/MS) (Fig 2A). MS/MS analysis of sample pre-enrichment was used to evaluate proteomic changes. Log10 transformed TMT ratios > 2 SD from the median, consistent in both clones, and within 95% of the WT2/WT1 confidence interval were considered significant. A total of 7930 unique phosphosites on 2,938 proteins were identified (Fig 2B; Dataset EV2). Using the defined selection criteria, we identified 41, 55, and 180 differentially regulated phosphosites in ΔT1, ΔT2, and ΔT3 cells, respectively (Fig 2B and C). The different *GALNT* isoform KOs each generated unique phosphorylation signatures associated with distinct cellular pathways, as shown by GO term enrichment analysis of significantly changed phosphoprotein identities (Fig 2D). First, ΔT1 exhibited phosphoproteomic changes associated with

cytoskeleton organization. For example, reduced phosphorylation of the important regulator of microtubule dynamics, stathmin at Ser63, suggests alterations in cell cycle progression through destabilization of microtubule assembly [38,39]. Other phosphorylation changes suggested an effect on cell cycle and proliferation, as well as terminal differentiation. Second, ΔT2 exhibited phosphoproteomic changes associated with cell adhesion and migration. For example, decreased phosphorylation at Ser418 of cortactin, a major protein involved in podosome formation, suggests impaired interaction with regulatory proteins required for cell motility [40]. Altered podosome formation may also affect extracellular matrix (ECM) remodeling and cell–matrix adhesion. In addition, ΔT2 was associated with changes in phosphorylation related to protein synthesis, such as several eukaryotic translation initiation factors, and other

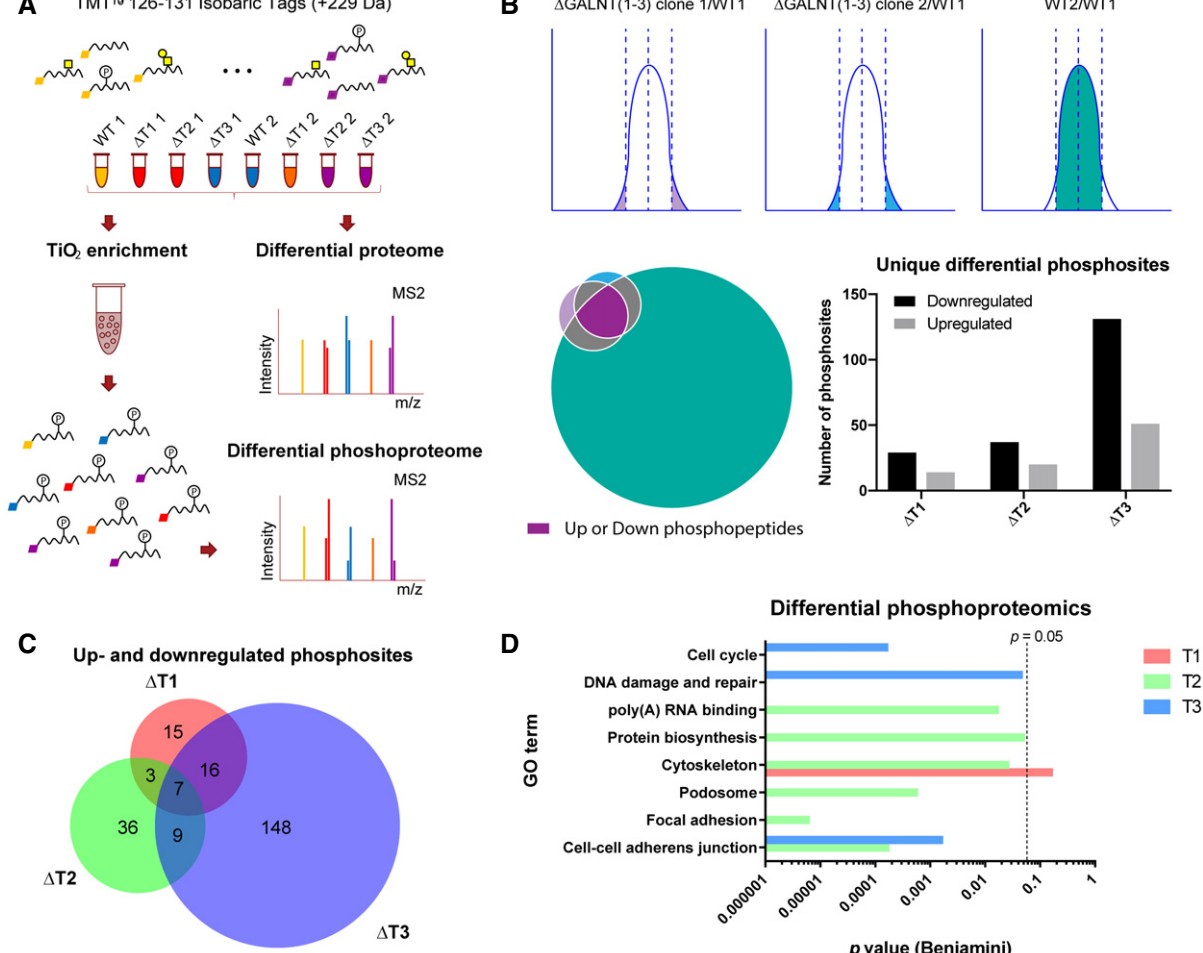

**Figure 2. Phosphoproteomic analysis.**

A   Strategy for differential phosphoproteomics of HaCaT *GALNT* KO cell lines, where the Gaussian distributions and the Venn diagram represent selection criteria.
B   Analysis of differentially phosphorylated peptides. Upregulated or downregulated phosphopeptides in *GALNT* KOs were considered significant, if phosphopeptides were identified in both clones more than 2× SD away from the median, and given the same phosphopeptides were within the normal variation of two wild-type biological replicates. The Venn diagram illustrates the selection criteria, where the color scheme corresponds to the Gaussian distribution illustration above. The bar graph depicts numbers of unique differential phosphosites in the individual *GALNT* KOs.
C   The Venn diagram represents overlap between differential phosphosites identified in the different *GALNT* isoform KOs.
D   GO terms (DAVID) associated with phosphoproteins identified in the 3 differential datasets. *P* value (Benjamini multiple comparison adjusted) of 0.05 is marked with a dashed line.

mRNA-binding proteins were identified as differential phosphotargets. Third, ΔT3 had the strongest effect on signaling networks and induced changes in phosphorylation associated with cell cycle progression, DNA repair, and protein degradation (Fig 2D; Dataset EV2). Importantly, the differentially phosphorylated targets in the *GALNT* KO cell lines were independent from changes in the proteome (Fig EV2), indicating that individual GalNAc-Ts control distinct biological pathways.

## Differential glycoproteomic analysis reveals that individual GalNAc-Ts control distinct subsets of protein targets

To investigate the molecular link between individual GalNAc-Ts and the distinct biological pathways affected by the loss of each transferase, we investigated which proteins were glycosylated by GalNAc-T1, GalNAc-T2, and GalNAc-T3, and where on the select protein targets the GalNAc/GalGalNAc residues were localized. We previously developed techniques that allow the identification of site-specific O-glycosylation on a proteome-wide basis, which has enabled high-throughput mapping of sites of O-glycan attachment in proteomes from both cell lines and tissues [28,30]. Furthermore, we have shown that targeted KO of individual *GALNT*s allows for discovery of GalNAc-T isoform-specific substrates [32–34,41]. Two approaches were used to characterize the differential O-glycoproteomes of ΔT1, ΔT2, and ΔT3. Initially, for ease of analysis, stable isotope dimethyl-labeled tryptic digests of isogenic cell pairs on a HaCaT *COSMC* KO background were coupled to VVA enrichment of desialylated glycopeptides (Fig 3A). Dimethyl labeling is a rather inexpensive method for quantitative proteomics but has a few limitations, such as non-identical retention times due to deuterium shift and only three different quantification channels [42]. We subsequently analyzed the effect of *GALNT* KOs on the more biologically relevant WT background. We have previously demonstrated efficient enrichment of desialylated ST structures by PNA/Jacalin lectin chromatography in epithelial cell lines and clinical specimens [28,43], and to allow side by side comparison of differentially modified targets in different cell lines, we used TMT labeling for this set of experiments [44]. TMT 10plex-labeled tryptic digests of three individual clones of each ΔT1, ΔT2, and ΔT3 were compared to WT in a single Jacalin LWAC run (Fig 3A). Only the TMT ratios consistently changed in all three clones were considered significant (Table 1; Dataset EV4). Each of the methods identified distinct subsets of GalNAc-T1, GalNAc-T2, and GalNAc-T3-specific targets (Fig 3B; Dataset EV4). An assessment of the isoform-specific glycosites identified in the WT cell background by the TMT method revealed 45 GalNAc-T1 sites, 40 GalNAc-T2 sites, and 34 GalNAc-T3 sites in glycopeptides carrying single HexHexNAc residues (Fig 3C). Considering glycosites on glycopeptides containing multiple HexHexNAc residues, the numbers of identified sites increased to 69 GalNAc-T1 sites, 51 GalNAc-T2 sites, and 56 GalNAc-T3 sites (Fig 3C). An independent analysis using Qlucore Omics Explorer software highlighted some of the same isoform-specific targets that contributed to the overall variation in the data (Fig 3E). The majority of isoform-specific glycosylation sites were not affected by knocking out alternative isoforms, but 3% of the glycosites were affected by the loss of each of the three enzymes. Similarly, approximately 15% of glycosites were affected by at least two isoforms (Fig 3D). Interestingly, 26% of GalNAc-T2 single sites were found in

other published and unpublished datasets in HepG2 cells and HEK293 cells, where GalNAc-T2 represents one of the predominant isoforms [32,34]. Lower consistency was observed for GalNAc-T1 or GalNAc-T3 single-site targets, which can partially be explained by a larger proportion of glycosites residing on peptides with multiple glycosites. Regardless, some of the GalNAc-T1 and GalNAc-T3 targets fit well with those identified previously in HepG2 and LS174 cells, respectively [32,33]. A total of 47, 82, and 82 glycosites glycosylated by GalNAc-T1, GalNAc-T2, and GalNAc-T3, respectively, on peptides with single or multiple HexNAcs were identified using the dimethyl labeling approach, with little overlap with the TMT dataset (Fig 3C). Some correlations were found with previously published differential glycoproteomic datasets on the *COSMC* KO background [32,33].

In conclusion, we unambiguously identified 67, 88, and 63 single-GalNAc/GalGalNAc glycosites for GalNAc-T1, GalNAc-T2, and GalNAc-T3, respectively. We found limited overlap between the identified glycosites between each GalNAc-T isoform, supporting the validity of the strategy and identified candidates. Comparison of glycoprotein and glycosite identities for each isoform between the two methods has shown variable overlap (Appendix Fig S1, Dataset EV4). Almost a third of glycoprotein substrates for GalNAc-T2 and GalNAc-T3 identified with TMT labeling were also identified with dimethyl labeling, whereas very little overlap was observed for GalNAc-T1 (Appendix Fig S1). However, the majority of commonly identified protein substrates were glycosylated at different positions using the two methods. This is not surprising given the different genetic backgrounds of the cells used for the analysis, as well as the distinct methodological approaches affecting the chemical properties and the dynamic range of detected glycopeptides. Transcriptomic and proteomic differences from WT cells should also be taken into account. Finally, it should be mentioned that similar to dimethyl labeling, the TMT labeling has its limitations, such as protease coverage, likely leading to underrepresentation of juxtamembrane sites and potential bias by select lectin enrichment of heavily glycosylated peptides.

Next, we performed GO term enrichment analysis on distinct subsets of targets identified for GalNAc-T1, GalNAc-T2, and GalNAc-T3. As expected, all three isoform-specific targets shared GO terms associated with the general properties of glycoproteins, such as signal peptide and transmembrane structural elements, as well as cell membrane localization or secretion from the cell (Fig 3F). In addition, each set of isoform-specific targets exhibited associations with distinct cellular processes (Fig 3G), with GalNAc-T1 targets associated with components of the endomembrane system, GalNAc-T2 targets associated with cell–ECM adhesion, and GalNAc-T3 targets associated with epithelial differentiation and cell migration. A few observations emerge based on the location of the top five downregulated single HexHexNAc glycosites found in individual *GALNT* isoform KOs. GalNAc-T1 and GalNAc-T2 glycosites were positioned mostly in unstructured regions, where GalNAc-T2-specific sites were commonly found close to protease cleavage sites. In contrast, the main GalNAc-T3 target glycosites resided on structural domains (Fig 3H). It is important to consider other factors that could play a role in defining enzyme preferences, including subcompartmentalization of the secretory pathway and complex formation between glycosyltransferases [45,46]. Furthermore, deletion or overexpression of individual isoforms will affect the overall

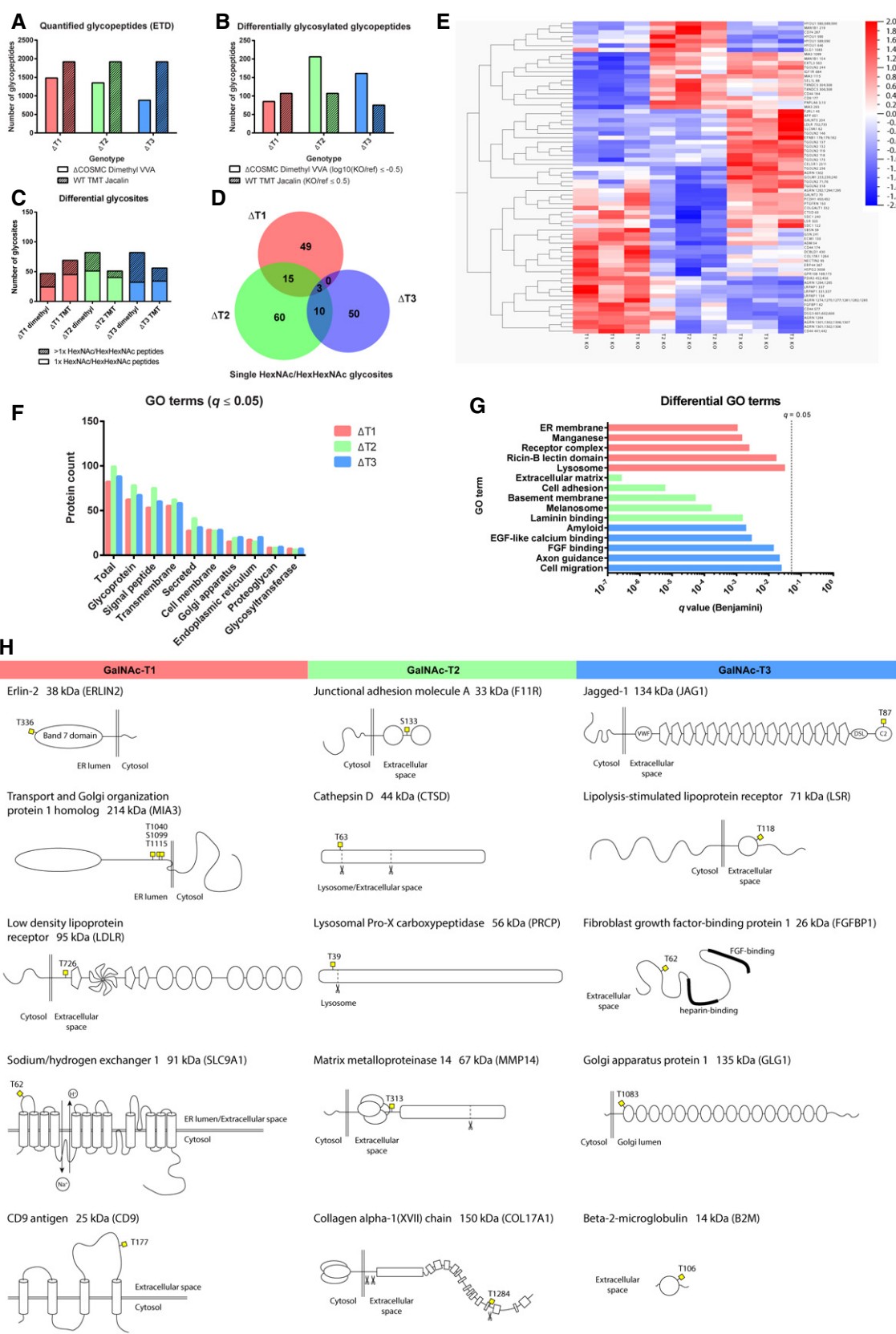

Figure 3.

Figure 3. O-glycoproteomic analysis.

A   Numbers of quantified O-glycopeptides identified with ETD fragmentation with either dimethyl labeling or TMT labeling approach in *COSMC* KO or WT background, respectively.

B   Numbers of differentially glycosylated peptides in the two approaches, using select filters on quantification values.

C   Numbers of unambiguous unique differential O-glycosites identified with the two different approaches. The stacked bar graph differentiates between glycosites identified on peptides with single and multiple HexNAc/HexHexNAc modifications.

D   The Venn diagram represents overlap between differential single HexNAc/HexHexNAc glycosites in the three differential datasets (pooled targets from both approaches).

E   Analysis of differential glycosites identified with the TMT labeling approach using the Qluqore Omics Explorer software. The heatmap represents O-glycosites most significantly contributing to sample variation, using ANOVA with a *q* value cutoff of 0.1, where each compared sample consists of three individual KO clones. Each horizontal line depicts the median quantification of glycopeptides contributing to the same glycosite. Alternative differentially glycosylated site positions are separated by semicolons on sites derived from glycopeptides with multiple HexNAc/HexHexNAc modifications.

F, G   GO term analysis of differentially glycosylated protein identifications from both approaches (DAVID, *q* ≤ 0.05, Benjamini adjusted). (F) GO terms found in all differential datasets associated with subcellular location and common glycoprotein properties. (G) Differential GO terms, found in only one of the three differential datasets. *P* value (Benjamini multiple comparison adjusted) of 0.05 is marked with a dashed line.

H   Top downregulated unambiguous single HexNAc/HexHexNAc targets found in all three clones of individual *GALNT* KO (not changed in other isoforms) with the TMT labeling approach.

O-glycosylation network of initiating and follow-up enzymes, likely causing changes in the O-glycoproteomic landscape [47].

## GalNAc-T1 is associated with integrity of the endomembrane system in human keratinocytes

The GalNAc-T1-specific targets included several ion antiporters crucial for maintaining intracellular pH and $Ca^{2+}$ homeostasis (SLC9A1, TMEM165; Fig 3H; Dataset EV4) [48–50]. Importantly, SLC9A1, the plasma membrane $Na^+/H^+$ exchanger NHE1, was identified as a differentially phosphorylated target, with the exact phosphosite known to regulate the activity of the protein (Dataset EV2) [51]. Intracellular pH measurements under steady-state conditions indicated that ΔT1 cells maintained a lower intracellular pH than WT (Fig 4D). Another target, TMEM165, functions as a calcium/proton transporter and is involved in calcium and lysosomal pH homeostasis [49]. A large number of targets associated with endomembrane compartments urged us to look at the prevalence of various intracellular organelles in ΔT1. Differences were observed in the intracellular trafficking compartments, with altered clathrin (Fig 4A) and Rab5 (Fig 4B) distribution and reduced numbers of recycling endosomes visualized by the loss of Rab11 staining (Fig 4C). In addition, we observed GalNAc-T1-selective glycosylation of several targets involved in skin differentiation and homeostasis, including a few recycling receptors. This includes the low-density lipoprotein (LDL) receptor, transforming growth factor β receptor 2 (TGFBR2), and insulin-like growth factor 1 receptor (IGF1R), and O-glycans have been suggested to play a functional role in both IGF1R signaling and LDL receptor function [52,53]. Mutation of the glycosylated amino acid on LDLR (Thr726Ile) has been identified in familial hypercholesterolemia and shown to reduce receptor activity to less than 5% in *ex vivo* assays [54]. Therefore, we tested receptor activity in ΔT1 by monitoring the uptake of pHrodo-labeled LDL, which fluoresces when taken up into endosomes/lysosomes. The uptake pattern in ΔT1 differed from the pattern in WT, with more dispersed localization of pHrodo-labeled LDL in the ΔT1 cells and lower grade of co-localization in lysosomes (Fig EV3). In general, ΔT1 cells poorly engaged in lysosome formation even 2.5 h after adding the labeled LDL. However, we cannot exclude that the pattern we observed is a result of alterations of the trafficking system. Furthermore, we

found downregulation of genes involved in cholesterol synthesis in ΔT1 (Dataset EV1). Given that LDL receptor is part of a feedback system that regulates cholesterol homeostasis, these results could suggest future exploration of site-specific glycosylation as a regulator of LDLR function.

Integration of the glycoproteomics, proteomics, and phosphoproteomics data supported the proposed role of GalNAc-T1 on the endomembrane system. It is however, important to note, that an alternative explanation for the quantitative enrichment of endomembrane compartment proteins in ΔT1 cells could be that it is a secondary effect of differentiation. Increased cell size and structural reorganization of the secretory pathway and vesicular transport during differentiation has previously been suggested for keratinocytes [55] and would fit well with the upregulation of differentiation markers and decreased cytosolic pH.

## GalNAc-T2 regulates cell–ECM adhesion in human keratinocytes

Among the identified GalNAc-T2-specific targets, we found several adhesion molecules (NRP2, F11R, PODXL, NECTIN2, CD99, PTPRS, BCAM, ITGAX, ROBO1, SEMA4D) and components of the basal lamina and ECM, including MATN2, COL4A1, ADAMTSL4, NID1, MMP14, ECM1, MMP1, LGALS3BP, COL12A1, CTSD, LAD1, VWA1, ECM1, and COL17A1 (Fig 3G; Dataset EV4). Immunofluorescence staining of components of different cell adhesion elements, including those of hemidesmosomes, desmosomes, adherens junctions, and tight junctions, revealed altered expression patterns for integrin alpha 6, as well as tight junction markers claudin 1 and occludin (Fig 4I, lower panel; Fig EV4). Furthermore, transmission electron microscopy (TEM) of organotypic tissue sections revealed reduced tight junction content in the ΔT2 organotypic skin (Fig 4I, upper panel). JAM1 (F11R), a component of tight junctions, was identified as a selective GalNAc-T2 target (Fig 3H; Table 1; Dataset EV4), but whether it has a direct impact on tight junction formation is unclear (Fig EV4). No change in surface expression was observed for integrin beta 4, integrin alpha 3, desmoglein 1, desmocollin 2, CD44, or collagen 17 α1 (Fig EV4). Together with the O-glycoproteomic data, these findings suggest that GalNAc-T2 function is critical for selective adhesion components including those important for cell–ECM interactions. To test this hypothesis, we evaluated the adhesion of ΔT2 cells to different

**Table 1. Isoform-selective glycosites.**

**WT TMT Jacalin (ΔT1/wt ≤ 0.5; ΔT2/wt > 0.5; ΔT3/wt > 0.5 (in all 3 clones))**

| Gene | Glycosite[a] | Protein name | Subcellular location | Function | Protein quant[b] (ΔT1/wt) | # of peptides | Coverage, % |
|---|---|---|---|---|---|---|---|
| FLRT3 | 402; 403 | Leucine-rich repeat transmembrane protein FLRT3 | ER, CM, Secr | Development | 1.097 | 5 | 10.79 |
| LDLR | 726 | Low-density lipoprotein receptor | CM, Endosome, Lysosome | LDL uptake | 0.840 | 5 | 6.05 |
| ERLIN2 | 336 | ER lipid raft associated 2 | ER | ERAD | 1.108 | 11 | 38.35 |
| CD9 | 177 | CD9 antigen | CM | Platelet activation | 1.047 | 5 | 16.23 |
| EXTL3 | 563 | Exostosin-like 3 | ER, Golgi | HS biosynthesis | 0.976 | 4 | 5.22 |
| SEL1L | 88 | Protein sel-1 homolog 1 | ER | ERAD/ERQC | 0.910 | 14 | 23.68 |
| GALNT5 | 179 | Polypeptide GalNAc transferase 5 | Golgi | O-glycosylation | 0.775 | 8 | 9.57 |
| MIA3 | 1,099 | Transport and Golgi organization protein 1 homolog | ER | Cargo loading | 0.762 | 30 | 21.19 |
| ST6GAL1 | 65; 66 | ST6 beta-galactosamide alpha-2,6-sialyltranferase 1 | Golgi | Glycosylation | | 1 or ND | |
| MAN1B1 | 154 | ER alpha-1,2-mannosidase | ER | ERQC | 0.949 | 9 | 19.89 |
| TGOLN2 | 202 | Trans-Golgi network integral membrane protein 2 | TGN, CM | Vesicular transport | 0.884 | 12 | 26.67 |
| IGFBP7 | 230; 239 | Insulin-like growth factor binding protein 7 | Secr | IGF binding | | 1 or ND | |
| IGF1R | 684 | Insulin-like growth factor 1 receptor | CM | Survival (RTK) | 1.146 | 16 | 14.12 |
| NUCB1 | 443 | Nucleobindin 1 | Golgi, Secr | IGF transport | 0.953 | 19 | 44.9 |
| STIM1 | 34 | Stromal interaction molecule 1 | ER, CM | Ca(2+) sensing | 0.784 | 18 | 32.85 |
| MIA3 | 293 | Transport and Golgi organization protein 1 homolog | ER | Cargo loading | 0.762 | 30 | 21.19 |
| PNPLA6 | 3; 10 | Patatin-like phospholipase domain containing 6 | ER | PtdCho metabolism | 1.014 | 11 | 9.59 |
| TGFBR2 | 39 | Transforming growth factor, beta receptor type II | CM | Cell signaling | | 1 or ND | |
| CA9 | 115 | Carbonic anhydrase 9 | CM | pH regulation | | 1 or ND | |
| RTN4RL1 | 321 | Reticulon 4 receptor-like 1 | CM | Receptor | 0.997 | 9 | 24.75 |
| CKAP4 | 471 | Cytoskeleton-associated protein 4 | ER, CM | ER anchoring | 0.769 | 34 | 21.9 |

**WT TMT Jacalin (ΔT2/wt ≤ 0.5; ΔT1/wt > 0.5; ΔT3/wt > 0.5 (in all 3 clones))**

| Gene | Glycosite | Protein name | Subcellular location | Function | Protein quant (ΔT2/wt) | # of peptides | Coverage, % |
|---|---|---|---|---|---|---|---|
| F11R | 133 | Junctional adhesion molecule A | CM, tight junction | Cell adhesion | 0.846 | 12 | 47.49 |
| TACSTD2 | 93 | Tumor-associated calcium signal transducer 2 | CM | Putative growth factor receptor | 1.075 | 14 | 38.39 |
| PRCP | 39 | Lysosomal Pro-X carboxypeptidase | Lysosome | Protease | 1.146 | 7 | 16.94 |
| CTSD | 63 | Cathepsin D | Secr, Lysosome | Protease | 0.956 | 26 | 62.62 |
| MMP14 | 313 | Matrix metallopeptidase 14 | CM | Protease | 1.044 | 18 | 32.82 |
| OS9 | 529 | Osteosarcoma amplified 9, ER lectin | ER | ERAD/ERQC | 0.996 | 6 | 11.39 |
| GPC1 | 510; 522 | Glypican 1 | CM, ECM, Endosome | HSPG | 0.975 | 16 | 38.17 |

**Table 1.** (continued)

| WT TMT Jacalin ($\Delta T2/wt \leq 0.5$; $\Delta T1/wt > 0.5$; $\Delta T3/wt > 0.5$ (in all 3 clones)) | | | | | | | |
|---|---|---|---|---|---|---|---|
| Gene | Glycosite | Protein name | Subcellular location | Function | Protein quant ($\Delta T2/wt$) | # of peptides | Coverage, % |
| TMTC3 | 831 | Transmembrane and tetratricopeptide repeat containing 3 | ER | Glycosylation | 1.035 | 15 | 19.56 |
| NUCB2 | 408 | Nucleobindin 2 | Secr, Golgi | Ca(2+) binding | 0.774 | 11 | 30.24 |
| COL17A1 | 1284 | Collagen, type XVII, alpha 1 | Basement M | Cell–matrix adhesion | 0.930 | 28 | 21.24 |
| HS2ST1 | 297 | Heparan sulfate 2-O-sulfotransferase 1 | Golgi | HS biosynthesis | 0.951 | 4 | 14.89 |
| NECTIN2 | 95 | Herpes virus entry mediator B | CM | Cell adhesion | | 1 or ND | |
| TGOLN2 | 318 | Trans-Golgi network integral membrane protein 2 | TGN, CM | Vesicular transport | 1.044 | 12 | 26.67 |

| WT TMT Jacalin ($\Delta T3/wt \leq 0.5$; $\Delta T1/wt > 0.5$; $\Delta T2/wt > 0.5$ (in all 3 clones)) | | | | | | | |
|---|---|---|---|---|---|---|---|
| Gene | Glycosite | Protein name | Subcellular location | Function | Protein quant ($\Delta T3/wt$) | # of peptides | Coverage, % |
| JAG1 | 87 | Jagged 1 | CM | Differentiation (Notch ligand) | 1.287 | 11 | 10.51 |
| DSG3 | 601; 602; 606 | Desmoglein 3 | CM, desmosome | Cell adhesion | 1.621 | 29 | 36.04 |
| CELSR1 | 2306; 2307; 2310; 2311 | Cadherin, EGF LAG seven-pass G-type receptor 1 | CM | Cell signaling | 1.163 | 7 | 3.65 |
| AGRN | 1294; 1295 | Agrin | CM, ECM | HSPG | 1.051 | 24 | 16.14 |
| FGFBP1 | 62 | Fibroblast growth factor binding protein 1 | CM, ECM | Mitogenic carrier protein | 1.506 | 4 | 24.79 |
| LSR | 118 | Lipolysis stimulated lipoprotein receptor | CM | Lipoprotein binding | 0.855 | 12 | 22.8 |
| GALNT5 | 260 | Polypeptide GalNAc-transferase 5 | Golgi | O-glycosylation | 1.017 | 8 | 9.57 |
| TGOLN2 | 174;175 | Trans-Golgi network integral membrane protein 2 | TGN, CM | Vesicular transport | 1.110 | 12 | 26.67 |
| FLRT3 | 397; 402; 405; 407 | Leucine-rich repeat transmembrane protein FLRT3 | ER, CM, Secr | Development | 1.294 | 5 | 10.79 |
| LRPAP1 | 134 | Alpha-2-macroglobulin receptor-associated protein | ER, Golgi, Endosome | Chaperone for LDLR related protein | 1.083 | 18 | 40.34 |
| GLG1 | 1083 | Golgi apparatus protein 1 | Golgi | FGF binding | 0.951 | 33 | 31.81 |
| B2M | 106 | Beta-2-microglobulin | Secr | Antigen presentation | 0.841 | 10 | 48.74 |
| CD59 | 76 | CD59 molecule, complement regulatory protein | CM, Secr | Complement binding | 0.869 | 6 | 35.94 |
| HYOU1 | 589; 590 | Hypoxia upregulated protein 1 | ER | Cytoprotection | 0.962 | 37 | 43.54 |
| TACSTD2 | 88 | Tumor-associated calcium signal transducer 2 | CM | Putative growth factor receptor | 0.974 | 14 | 38.39 |
| FAT2 | 4036 | Protocadherin Fat 2 | CM, cell junction | Cell migration | 1.034 | 63 | 19.54 |

CM, cell membrane; ER, endoplasmic reticulum; ERAD, ER-associated protein degradation; ERQC, ER quality control; HSPG, heparan sulfate proteoglycan; ND, not detected; Secr, secreted; TGN, trans-Golgi network.
[a]Semicolons separate alternative differentially glycosylated site positions (derived from glycopeptides with multiple HexNAc/HexHexNAc modifications).
[b]Average of three clone ratios (KO/wt); filtered for values > 0.75.

ECM components and found that $\Delta T2$ cells exhibited a weaker association with fibronectin, laminin, collagen 1, and fibrinogen than WT cells (Fig 4E).

As seen in other cell systems [41], many of the GalNAc-T2 glyco-sites were found to reside in protein regions associated with proteolytic processing (CTSD, PRCP, COL17A1, BCAM, LRP1) (Fig 3H; Dataset EV4). One such glycosite is residing at the fourth non-collagenous domain of COL17A1 cleaved by plasmin and harboring a few known mutations in skin blistering diseases [56,57]. Lack of protection due to the loss of a glycosylation site

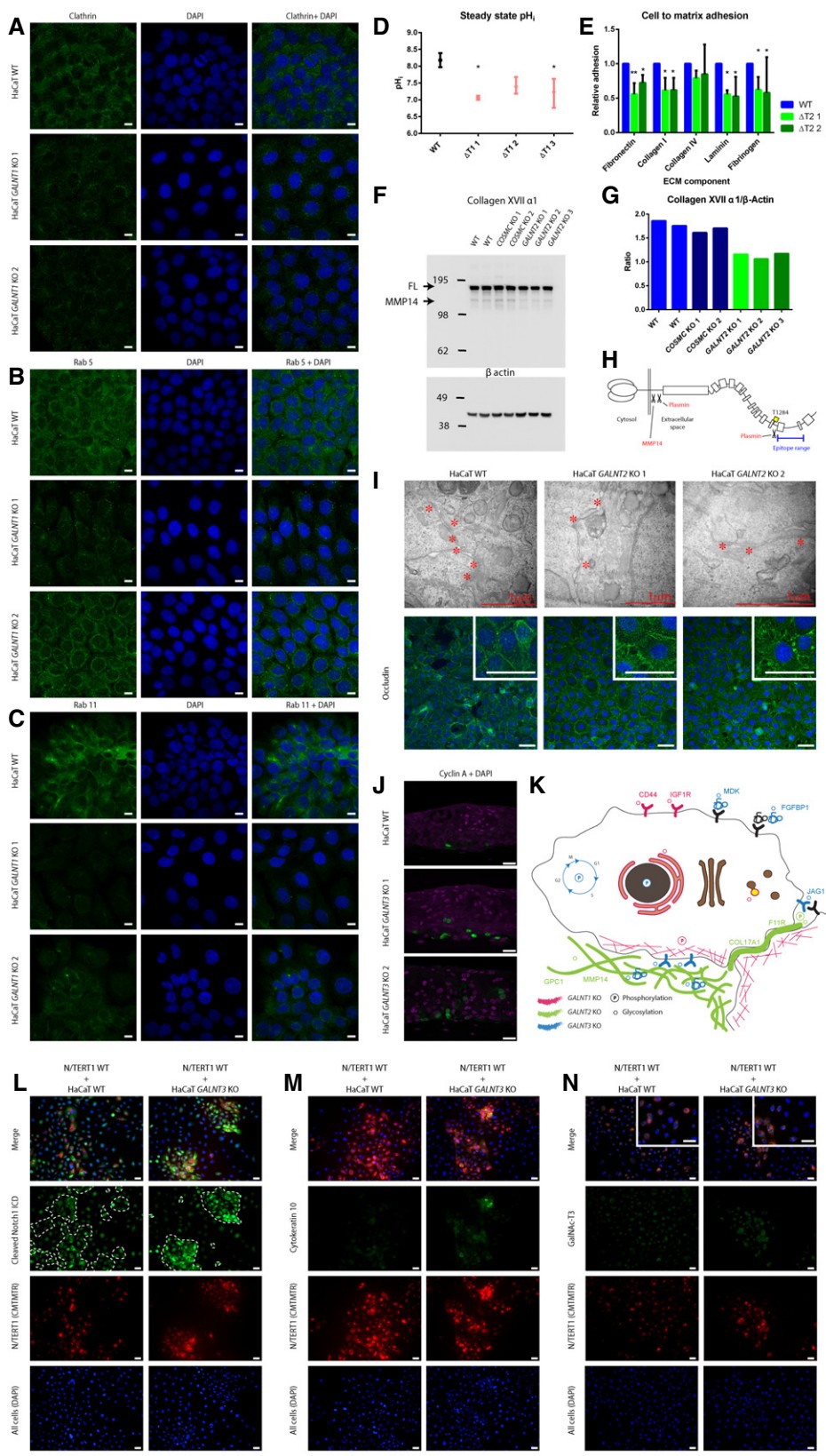

**Figure 4.**

**Figure 4. Pathways affected by isoform-specific O-glycosylation.**

A–D  GalNAc-T1 affects the organization and function of the endomembrane system. HaCaT WT and *GALNT1* KO cells were stained for intracellular trafficking markers (scale bar—10 μm). (A) Clathrin; (B) Rab5; and (C) Rab11. (D) Steady-state intracellular pH measurement in in HaCaT WT and *GALNT1* KO cells. Data are presented as average ± SD of 2 independent experiments. ANOVA followed by Dunnet's multiple comparison test was used to compare mean intracellular pH of three different clones of *GALNT1* KOs to WT. *$P \leq 0.05$.

E  GalNAc-T2 is important for cell to matrix adhesion. HaCaT WT and *GALNT2* KO cells were used in a cell to matrix adhesion assay using plates coated with different ECM components. Data are presented as average + SD of three independent experiments and are normalized to WT. ANOVA followed by Dunnet's multiple comparison test was used to compare mean relative adhesion of two different clones of *GALNT2* KOs to WT. *$P \leq 0.05$; **$P \leq 0.01$.

F  Western blot for GalNAc-T2-specific target and hemidesmosome component COL17A1.

G  Quantification of Western blot bands for COL17A1 (full length) normalized to β actin.

H  Depiction of COL17A1 protein, including position of GalNAc-T2-specific O-glycosite and proteolytic processing sites.

I  GalNAc-T2 may affect tight junction formation and integrity. Upper panel, TEM of WT and *GALNT2* KO organotypic tissue sections, where tight junctions are marked with asterisks. Lower panel, IF staining of WT and *GALNT2* KO for tight junction marker occludin (scale bar—50 μm). Inserts depict 3× zoomed in partial images (scale bar—50 μm).

J  Expression of cyclin A (green) in HaCaT WT and *GALNT3* KO organotypic tissue, visualized by IF (scale bar—20 μm).

K  The cartoon summarizes the effects of individual *GALNT* isoform KOs on signaling networks and subcellular location of protein substrates based on differential phosphoproteomic and O-glycoproteomic analyses.

L–N  Co-culture of HaCaT WT or *GALNT3* KO keratinocytes with N/TERT1 keratinocytes induces cellular differentiation. Cells co-cultured on coverslips were stained for cleaved Notch 1 intracellular domain (L), cytokeratin 10 (M), or GalNAc-T3 (N; inserts depict 2× zoomed in partial images (scale bar—20 μm)). Green—relevant staining, red—N/TERT1 cells (labeled with CMTMTR), and blue—DAPI, scale bar—20 μm. Boundaries between HaCaT and N/TERT-1 cells in the green panel of (L) are marked with a dashed line.

could enhance plasmin cleavage and partially explain the impaired cell to matrix adhesion in ΔT2 cells. In support of this, we observed decreased amounts of full-length and shed COL17A1 on Western blot (Fig 4F–H), but total protein levels were unchanged according to MS (Table 1, Dataset EV3 and EV4), suggesting that additional cleavage of the ectodomain may be enhanced in ΔT2.

Cleavage of the full-length and shed domain by plasmin results in disruption of the C-terminal region important for interaction with laminin 5 and creates neoepitopes for linear IgA dermatosis antibodies [58,59]. In this context, we found LAD1, another classical IgA dermatosis marker, as a GalNAc-T2 target. The basal cell adhesion molecule (BCAM) was also among the selective targets with the glycosylation site in close proximity to protease cleavage sites. It is tempting to speculate that GalNAc-T2 could affect BCAM shedding known to modulate interaction of BCAM with laminin 5 [60,61]. Moreover, an SNP in the glycosylated Ser539 of BCAM differentiates two different Lutheran blood group antigens [62]. In conclusion, these results suggest that GalNAc-T2 impacts the stability of cell–matrix adhesion, correlating well with the basal expression pattern of GalNAc-T2 in both the epidermis and several other epithelia.

## GalNAc-T3 regulates epithelial differentiation in human keratinocytes

Among the GalNAc-T3-specific targets, we found an enrichment of proteins containing EGF-like $Ca^{2+}$ binding domains, as well as some growth factor binding and heparin binding proteins (Fig 3G; Dataset EV4). Interestingly, we found specific sites in the Notch ligand Jagged-1, the adhesive proteins, protocadherin Fat 2, and desmoglein 3, which are involved in cellular differentiation and proliferation (Table 1, Dataset EV4). Consistent with delayed differentiation in the ΔT3 skin, we detected increased expression of cyclin A in the K10-negative skin layers (Fig 4J). The GalNAc-T3-specific glycosylation on position T87 in Jagged-1 could potentially explain the prominent altered differentiation phenotype observed in ΔT3 skin. Interestingly, Jagged-1 harboring a mutation at Thr87 has been reported to have increased affinity for the Notch1 receptor through fixation of the catch-bond interaction [63]. In order to address the

potency of Jagged-1^T3KO to induce Notch1 signaling, we co-cultured HaCaT WT or ΔT3 keratinocytes with N/TERT1 immortalized keratinocytes and assessed Notch1 activation by nuclear expression of cleaved Notch1 intracellular domain (NICD). In addition, we evaluated differentiation by K10 staining. Co-culture of N/TERT1 WT and HaCaT ΔT3 cells induced robust differentiation of N/TERT1 cells, which were mostly organized into compact piles, strongly positive for nuclear cleaved NICD, and increased K10 expression (Fig 4L–N). In contrast, co-culture of WT HaCaT and N/TERT1 keratinocytes only induced limited K10 expression and failed to induce organized piles of keratinocytes positive for NICD (Fig 4L–N). Furthermore, a lower level of Notch1 intracellular domain cleavage was observed in HaCaT WT, with fewer cells positive for cleaved NICD in HaCaT ΔT3 cells. The combined results support the potential role of GalNAc-T3-specific O-glycosylation on differentiation possibly through fine-tuning Jagged–Notch interactions. We also investigated Notch1 signaling in the organotypic models by visualizing localization of cleaved NICD, which was primarily detectable in the nuclei of suprabasal cell layers of WT HaCaT skin (Fig EV5). Compared to WT, little Notch1 signaling was observed in suprabasal layers of ΔT3 organotypic models (Fig EV5). Whether the observed GalNAc-T3 glycosylation of Jagged-1 is the direct cause of the perturbed differentiation observed in the ΔT3 skin equivalents is unclear. As an alternative explanation, loss of GalNAc-T3 may affect the intercellular adhesion system by specific glycosylation of both protocadherin Fat 2 and desmoglein 3, which are involved in cell migration and cell–cell adhesion, respectively. Furthermore, a number of FGF and heparin binding proteins (FGFBP1, GLG1, GPC1) that modulate the availability of these growth factors have been identified as GalNAc-T3-specific targets and may contribute to the decreased differentiation of suprabasal keratinocytes via FGFR signaling [64].

Given that only a few overt phenotypes present in animal models and humans are associated with loss of GalNAc-T function, it is in some ways surprising that loss of single GalNAc-Ts causes such distinct phenotypes in the organotypic skin models. Loss of GalNAc-T3 in humans seems to primarily affect phosphate homeostasis in familial tumoral calcinosis [13,4,65]. Different, more subtle phenotypes include a strong genome-wide association of GalNAc-T2 with

plasma lipid profiles [21,26,66] or GalNAc-T11 with congenital heart disease [19]. In *GALNT1* KO mice, several different issues have been observed, including a bleeding disorder, decreased leukocyte homing capacity, and impaired heart valve development [67–69], though similar associations have not yet been discovered in humans. In all cases, the results suggest that it is only a few isoform-specific GalNAc-T targets that cause the observed phenotypes, and that the other isoforms and feedback mechanisms are able to compensate to a large extent. In this context, certain stress challenges are likely needed in model systems to evoke more overt phenotypes caused by the lack of a single GalNAc-T. Thus, it is conceivable that the precancerous nature of the HaCaT cells used in the present organotypic skin model only requires minimal perturbations to provoke the overt phenotypes observed in *GALNT1* KO, *GALNT2* KO, and *GALNT3* KO tissues.

In summary, we show that O-glycosylation is a widespread modification of proteins expressed in human epithelial cells. Furthermore, our results provide evidence that the O-glycosylation landscape is controlled by the expressed repertoire of GalNAc-Ts, each with specific but overlapping substrate specificities directing distinct biological functions in epithelial homeostasis. Finally, we think that the organotypic model presented here is an enabling strategy for studies of site-specific glycosylation using precise genome editing techniques combined with sophisticated MS.

# Materials and Methods

### Tissues

Human skin was obtained from Bodylift Surgical Clinic with approval from the Regional Ethical Committee. Sections (5 μm thick) were cut from frozen blocks and mounted on gelatin-coated slides. Every fifth section was stained with hematoxylin–eosin and used as a reference during evaluation. Sections were fixed in cold 10% buffered neutral formalin for 15 min or cold acetone for 10 min. Immunohistochemistry (IHC) was performed as described below.

### Cell culture

The HaCaT human keratinocyte cell line was grown in DMEM supplemented with 10% FBS (HyClone) and 4 mM L-glutamine. N/TERT-1 immortalized keratinocytes (obtained from JG Rheinwalds lab, Harvard Institute of Medicine, Brigham & Women's Hospital) were cultured in keratinocyte serum-free medium (K-SFM, Gibco, Thermo Scientific) with 12.5 mg bovine pituitary extract per 500 ml medium (Gibco, Thermo Scientific), 0.2 ng/ml epidermal growth factor (EGF) (Thermo Scientific), and CaCl$_2$ (Sigma) to a final Ca$^{2+}$ concentration of 0.3 mmol/l, as described previously [70]. Prior to staining, cells were trypsinized and seeded on diagnostic slides, dried overnight, and fixed in ice-cold acetone for 5 min. Alternatively, cells were cultured directly on coverslips in 24-well plates and fixed directly in the wells with 4% PFA in PBS for 10 min, followed by permeabilization for 3 min with 0.3% Triton X-100 in PBS at room temperature (RT), or for 5 min on ice in ice-cold acetone/methanol (1:1). For co-culture experiments, N/TERT-1 keratinocytes were labeled with 5 μM CellTracker™ Orange CMTMR

Dye (Thermo Scientific) for 45 min at 37°C, then trypsinized, mixed with unlabeled HaCaT keratinocytes in a 1:1 ratio (5 × 10$^4$ cells of each), and seeded onto glass coverslips in complete K-SFM. Twenty-four hours later, the cells were shifted to low Ca$^{2+}$ medium (50% DMEM:F12, 50% K-SFM, supplemented with BPE and EGF) and cultured for additional 24 h before harvesting.

### ZFN and CRISPR/Cas9 gene targeting

HaCaT *GALNT1*, *GALNT2*, and *GALNT3* KOs on a WT and Simple-Cell (*COSMC* KO) background were generated as described previously [71]. Briefly, ZFN constructs targeting *COSMC*, *GALNT1*, *GALNT2*, and *GALNT3* were custom produced (Sigma-Aldrich) and HaCaT cells transfected with 2 μg of each GFP- and E2-crimson-tagged ZFN plasmids [72] via nucleofection with Amaxa Nucleofector (Lonza). Co-transfection of non-tagged ZFN constructs (2 μg of each) and 0.3 nmol of single-stranded oligonucleotide (ssODN) was used to generate some of the *GALNT3* KO clones. GFP and E2-crimson double-positive cells were enriched by fluorescence-activated cell sorting (FACS) and single cells cloned by limited dilution. Indels at the respective target sites were characterized by Indel Detection by Amplicon Analysis (IDAA) [73] and monoclonal antibody staining. Indels identified in individual cell clones were confirmed by Sanger sequencing. Additional *GALNT3* KO clones were generated by co-transfecting 3.5 μg of each CRISPR/Cas9-GFP and a gRNA construct targeting a different exon than the ZFN nucleases. GFP-positive cells were enriched by FACS and single cells cloned, followed by validation with mAb staining and Sanger sequencing.

### Raft culture

Organotypic cultures were prepared as described by Dabelsteen *et al* [29,74]. Briefly, human fibroblasts were suspended in acid-extracted type I collagen (4 mg/ml) and allowed to gel over a 1-ml layer of acellular collagen in six-well culture inserts with 3-μm-pore polycarbonate filters (BD Biosciences NJ, USA). Gels were allowed to contract for 4–5 days before seeding with 3 × 10$^5$ HaCaT keratinocytes in serum and mitogen-supplemented DMEM/F12 raft medium. Inserts were raised to the air–liquid interface 4 days after cell seeding, and the media changed every second day for an additional 10 days. At least three experiments were conducted at different time points with three different clones of each KO cell line, resulting in organotypic cultures with similar morphologies. The exception was the additional *GALNT3* KOs prepared with CRISPR/Cas9 targeting a different exon than the ZFN nucleases, which were only tested once. Organotypic sections were prepared and stained histochemically as described previously.

### Immunofluorescence

Slides, coverslips, or frozen tissue sections were incubated overnight at 4°C with undiluted hybridoma supernatants or lectins: anti-GalNAc-T1, mouse mAb 4D8; anti-GalNAc-T2, mouse mAb 4C4; anti-GalNAc-T3, mouse mAb 2D10; anti-Tn, mouse mAb 5F4; anti-STn, mouse mAb 3F1 or anti-T, biotinylated peanut agglutinin (Vector Labs, 0.5 μg/ml), ± neuraminidase treatment (*Clostridium perfringens* neuraminidase, 0.1 U/ml in 0.05 M NaAc for 1 h at 37°C); rabbit anti-clathrin (1:50), rabbit anti-Rab5 (1:100), rabbit

anti-Rab11 (1:100; all from CST, Endosomal Marker Antibody Sampler Kit); rabbit anti-claudin 1 (1:200, Proteintech); mouse anti-occludin (1:200, Proteintech); mouse anti-JAM-A (1:100, BIO-RAD); mouse anti-integrin alpha 3 (1:20 R&D Systems); mouse anti-integrin alpha 6 (1:50, BIO-RAD); rat anti-integrin beta 4 (1:200, LSBio); rabbit anti-collagen XVII α1 (1:200, Abcam); goat anti-desmoglein 1 (1:50, Proteintech); rabbit anti-desmocollin 2 (1:200, Proteintech); goat anti-E-cadherin (1:200, R&D Systems); mouse anti CD44 (1:500, R&D Systems); mouse anti-cytokeratin 10 (1:200, DAKO, Denmark); and rabbit anti-cleaved Notch1 ICD (1:100, CST). Bound mAbs were detected with FITC-conjugated rabbit anti-mouse immunoglobulins (1:100; DAKO, Denmark), rabbit F(ab')$_2$ anti-mouse IgG Alexa 488 (1:500), goat anti-mouse IgG Alexa 488 (1:500), goat anti-rabbit IgG Alexa 488 (1:500), donkey anti-goat IgG Alexa 488 (1:500), goat anti-rat IgG Alexa 488 (1:500), and streptavidin Alexa 488 (1:500) (all from Molecular Probes, Thermo Scientific, EU), or SuperBoost Tyramide kit (Thermo Fisher Scientific) according to the manufacturer's protocol. Slides and coverslips were mounted with ProLong Gold Antifade Reagent (with or without DAPI) (Molecular Probes, Thermo Scientific, EU). Coverslips were stained with 1 μg/ml DAPI solution for 3 min before mounting. Fluorescence micrographs were obtained on a Leica wide-field fluorescence microscope or a Zeiss LSM710 confocal microscope. Images were assembled using Adobe Photoshop CS6, Adobe Illustrator CS6, or Zeiss ZEN Lite software.

### Immunohistochemistry

Paraffin-embedded sections (3–5 μm) were used for IHC. Heat-induced antigen retrieval was performed in Tris-EDTA (pH 9) or citrate buffer (pH 6) before applying primary antibodies: mouse anti-cytokeratin 10 (1:100, DAKO), rabbit anti-cleaved Notch1 ICD (1:100, CST), mouse anti-cyclin A (1:100, Novocastra Laboratories), and mouse anti-Ki67 (1:100, DAKO). IHC was performed using the SuperBoost Tyramide kit (Thermo Fisher Scientific) or the Ultra Vision Quanto Detection System HRP DAB (Thermo Scientific) according to the manufacturers' protocols. Sections were mounted with ProLong Gold Antifade Reagent with DAPI (Molecular Probes, Thermo Scientific, EU) or Tissue-Tek Compound (Sakura).

### Quantification of organotypic cultures

The K10-negative area was quantified using AxioVision software. Epidermal thickness was measured on H&E images by making four vertical lines of equal distance. Five images were quantified per clone. The K10-negative area was determined by outlining the area that was not stained positive by an anti-K10 antibody in a total of 5 images per KO clone.

### Transmission electron microscopy

TEM was performed as described previously [29]. The samples were fixed with 2% (vol/vol) glutaraldehyde in 0.05 M sodium phosphate buffer (pH 7.2), rinsed three times in 0.15 M sodium cacodylate buffer (pH 7.2), and subsequently post-fixed in 1% (wt/vol) OsO$_4$ in 0.12 M sodium cacodylate buffer (pH 7.2) for 2 h. The specimens were dehydrated in a graded series of ethanol, transferred to propylene oxide, and embedded in Epon

according to standard procedures. Sections (< 80-nm-thick) were cut with a Reichert-Jung Ultracut E microtome and collected on one-hole copper grids with Formvar supporting membranes, stained with uranyl acetate and lead citrate, and subsequently examined with a Philips CM 100 transmission electron microscope (Philips, Eindhoven) operated at an accelerating voltage of 80 kV and equipped with an OSIS Veleta digital slow scan 2,000 × 2,000 CCD camera. Digital images were recorded using the ITEM software package.

### RNA transcriptomic analysis

Total RNA was extracted from cells 48 h post-confluency using the RNeasy® kit (Qiagen). RNA integrity and quality were determined using Bioanalyzer instrumentation (Agilent Technologies). Analyses were performed on total RNA from three clones of each GALNT1 KO, GALNT2 KO, and GALNT3 KO and two biological replicates of WT HaCaT cells. Transcriptome analysis of the extracted total RNA was performed by Beijing Genomics Institute (BGI) as reported previously [32]. Briefly, a library was constructed using the Illumina Truseq RNA Sample Preparation Kit and subjected to PCR amplification and quality control (QC) before undergoing next-generation sequencing with the Illumina HiSeq 2000 System (Illumina, USA).

### Bioinformatics analysis

Bioinformatics analysis was performed as described previously [32]. Briefly, aligned reads from the RNAseq analysis were analyzed using the DESeq [75] and EdgeR [76] packages for R and Bioconductor to identify differentially expressed transcripts. DESeq and EdgeR analyses were run using default parameters and following previously defined protocols [77]. Alternatively, single reads were aligned to the human genome hg19 reference sequence and analyzed using the CLC Genomics Workbench (Qiagen).

### TiO$_2$ enrichment of phosphopeptides

HaCaT ΔT1, ΔT2, ΔT3 (two clones of each) and two biological replicates of WT grown in culture dishes were harvested by scraping in modified RIPA buffer (50 mM Tris pH 7.5, 150 mM NaCl, 1% NP-40, 0.1% Na deoxycholate, 1 mM EDTA) supplemented with Complete Mini protease inhibitors and a mix of phosphatase inhibitors. Acetone-precipitated proteins were reconstituted in 0.1% RapiGest (Waters) in 50 mM ammonium bicarbonate, reduced (1 mM DTT 1 h RT) and alkylated (5.5 mM CAA 1 h RT), and treated overnight with PNGase F, followed by digestion with trypsin at RT for 16 h. A total of 200 μg of each digest was labeled using the TMT-10plex Kit (Thermo Scientific) according to the manufacturer's instructions, mixed in equimolar ratio, and enriched using TiO$_2$ beads [78]. Briefly, peptides were mixed with TiO$_2$ beads (0.6 mg for 100 μg peptides) in 1 M glycolic acid, 80% ACN, and 5% TFA, and vortexed vigorously for 15 min at RT. The procedure was repeated with half the amount of beads. The two batches of beads were washed with loading buffer, followed by 80% ACN, 1% TFA and 20% ACN, 0.2% TFA. The phosphopeptides were eluted with 1% NH$_4$OH. Peptides before enrichment and eluted phosphopeptides were orthogonally separated using high pH fractionation and analyzed by HCD MS.

## LWAC isolation of Tn and T-O-glycopeptides

LWAC isolation of Tn and T-O-glycopeptides was performed as described previously [27,30]. Briefly, proteins were reduced (5 mM DTT 45 min 60°C) and alkylated (10 mM IAA 30 min RT), digested with trypsin (Roche), and neuraminidase-treated to remove sialic acid residues. For experiments in *COSMC* KO (SC), these steps were followed by labeling with light or medium isotopomeric dimethyl labels [42]. The labeled digests were mixed in a 1:1 ratio in the following sets: HaCaT$^{SC}$ (light) and HaCaT$^{SC}$ΔT1 (medium), HaCaT$^{SC}$ (light) and HaCaT$^{SC}$ΔT2 (medium), and HaCaT$^{SC}$ (light) and HaCaT$^{SC}$ΔT3 (medium). The mixed pairs of digests were enriched using agarose-bound VVA LWAC and eluted with GalNAc. For experiments in WT, digests of three clones of each ΔT1, ΔT2, and ΔT3, and one WT were labeled using the TMT-10plex Kit (Thermo Scientific) according to the manufacturer's instructions. Labeled digests were mixed in an equimolar ratio and enriched using Jacalin LWAC, followed by elution with D-galactose. Isoelectric focusing was performed on VVA-enriched glycopeptides by a 3100 OFFGEL fractionator (Agilent) using pH 3–10 strips (GE Healthcare) [79]. High pH fractionation was performed on TMT-labeled Jacalin-enriched glycopeptides, as well as peptides before enrichment. All glycopeptide samples were desalted by self-made Stage Tips (C18 sorbent from Empore 3 M) and submitted to LC-MS and HCD/ETD-MS/MS.

## Mass spectrometry

EASY-nLC 1000 UHPLC (Thermo Scientific) interfaced via a Nanospray Flex ion source to an LTQ-Orbitrap Velos Pro or Orbitrap Fusion spectrometer (Thermo Scientific) was used for peptide, glycopeptide, and phosphopeptide analysis as described previously [27]. Data were processed using Proteome Discoverer 1.4 software (Thermo Scientific).

## Intracellular pH measurements

Intracellular pH measurements were performed as described previously [80]. Cells were seeded in WillCo glass-bottom dishes (WillCoWells) and loaded with 2′,7′-bis-(2-carboxyethyl)-5-(and-6)-carboxyfluorescein acetoxymethyl ester (BCECF-AM, 1.6 µM; Invitrogen) in growth medium for 30 min at 37°C prior to sample collection. Cells were washed once in isotonic Ringer solution and placed in a temperature-controlled compartment (37°C) equipped with gas and solute perfusion. The pHi measurements were carried out using a Nikon Eclipse Ti microscope equipped with a Xenon lamp for fluorescence excitation, a 40× oil 1.4 NA objective, and EasyRatioPro imaging software (PTI). Emission was measured at 520 nm after excitation at 440 nm and 485 nm. For acid loading, cells were exposed to saline supplemented with 20 mM NH$_4$Cl for 10 min. Isotonic calibration solutions were used for a 4-point linear calibration curve to calibrate the pHi values. Fluorescence measured from the two excitation channels (440 nm and 485 nm) was corrected for the respective background fluorescence, which was assessed by measuring a cell-free area during the experiment. The 485 nm/440 nm ratio was then calculated and the calibration data fitted to a linear function in the applied pH range, in which the experimental data were inserted and

thereby converted to corrected pH values. Average pHi measurements for each time point and their standard deviations were calculated as in Ref. [80].

## Protein extraction and Western blotting

Cells grown in cell culture dishes were washed with ice-cold PBS and lysed in modified RIPA buffer (50 mM Tris pH 7.5, 150 mM NaCl, 1% NP-40, 0.1% Na deoxycholate, 1 mM EDTA) supplemented with protease and phosphatase inhibitors, followed by sonication using a sonic probe on ice. Proteins were mixed with 4× NuPAGE sample buffer and 10 mM DTT, heat denatured (95°C 5 min), and separated on Novex 4–12% gradient gel (Bis-Tris) in 1× NuPAGE MES running buffer (Invitrogen), followed by transfer onto nitrocellulose membrane in 20% MeOH in running buffer at 320 mA overnight at 4 °C. Membranes were blocked with 5% skim milk in TBS-T and blotted with rabbit anti-collagen XVII α1 (1:1,000, Abcam) or mouse anti-β actin (1:10,000, Sigma) antibodies, followed by goat anti-rabbit Igs-HRP or rabbit anti-mouse Igs-HRP (1:4,000, DAKO). Membranes were developed using Pierce ECL or ECL Plus Kits (Thermo Scientific) and visualized using the ImageQuant LAS4000 system. Protein bands were quantified using ImageJ software.

## LDL uptake

Cells grown on coverslips were cultured in media supplemented with 10% lipoprotein-depleted serum (Sigma-Aldrich) for 24 h, followed by pulsing with 70 nM Lysotracker Red DND-99 (Thermo Fisher Scientific) (−1 h) and 14 µg/ml LDL-pHrodo (Thermo Fisher Scientific) (0 h). Coverslips were fixed with 4% PFA for 5 min at 1 h, 1.5 h, and 2.5 h and imaged using confocal microscopy.

## Cell–matrix adhesion assay

Cell to ECM adhesion was tested using the CytoSelect™ 48-well Cell Adhesion Assay (Cell Biolabs) according to the manufacturer's protocol. Briefly, the cells were trypsinized and allowed to adhere for 1 hour at 37°C. The unbound cells were washed off, stained, lysed, and quantified using a plate reader at a wavelength of 560 nm. The experiment was set up in duplicate for each cell line. The results are representative of three independent experiments.

# Data availability

The RNAseq data from this publication have been deposited to the Gene Expression Omnibus GEO database (https://www.ncbi.nlm.nih.gov/geo/query/acc.cgi?acc = GSE141387) and assigned the identifier GSE141387. The mass spectrometry-based data from this publication (phosphoproteomics, glycoproteomics, and proteomics) have been deposited to the ProteomeXchange Consortium via the PRIDE partner repository (https://www.ebi.ac.uk/pride/) and assigned the identifier PXD016618.

**Expanded View** for this article is available online.

## Acknowledgments

This work was supported by the European Commission (GlycoSkin H2020-ERC GAP-772735), the Danish National Research Foundation (DNRF107), The Friis Foundation, The Michelsen Foundation, The Danish Research Councils (Sapere Aude Research Leader grant to HW), Novo Nordisk Foundation, the Danish Strategic Research Council, the Lundbeck Foundation (R219-2016-545), and the program of excellence from the University of Copenhagen (CDO2016). We acknowledge the Core Facility for Integrated Microscopy, Faculty of Health and Medical Sciences, University of Copenhagen. We thank Lars Hansen (Faculty of Health and Medical Sciences, University of Copenhagen) for initial analysis of RNAseq data. We also thank professor Jesper Reibel (Faculty of Health and Medical Sciences, University of Copenhagen) for interpretation of glyco-engineered tissue morphology.

## Author Contributions

IB, SD, and HHW conceived and designed the study; IB, EMHP, ZY, SD, MIN, INM, SHK, HJJ, SFP, SYV, EPB, and HHW contributed with experimental data and interpretation; IB and HHW wrote the manuscript; and all authors edited and approved the final version.

## Conflict of interest

The authors declare that they have no conflict of interest.

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
