## [Review Process File · EMBO Reports]

O-glycan initiation directs distinct biological pathways and controls epithelial differentiation

Ieva Bagdonaite, Emil M. H. Pallesen, Zilu Ye, Sergey Y. Vakhrushev, Irina N. Marinova, Mathias I. Nielsen, Signe H. Kramer, Stine F. Pedersen, Hiren J. Joshi, Eric P. Bennett, Sally Dabelsteen, and Hans H. Wandall

Review timeline:

Submission date:	16 July 2019
Editorial Decision:	27 August 2019
Revision received:	6 December 2019
Editorial Decision:	14 February 2020
Revision received:	3 March 2020
Accepted:	16 March 2020

Transaction Report:

1st Editorial Decision

27 August 2019

Thank you for the submission of your research manuscript to EMBO Reports. We have now received the full set of referee reports that is copied below.

As you will see, the referees acknowledge that the findings are potentially interesting. However, referees 1 and 2 also point out several technical concerns and have a number of suggestions for how the study should be strengthened, and I think that all of them should be addressed.

Given these constructive comments, we would like to invite you to revise your manuscript with the understanding that the referee concerns (as detailed above and in their reports) must be fully addressed and their suggestions taken on board. Please address all referee concerns in a complete point-by-point response. Acceptance of the manuscript will depend on a positive outcome of a second round of review. It is EMBO reports policy to allow a single round of revision only and acceptance or rejection of the manuscript will therefore depend on the completeness of your responses included in the next, final version of the manuscript.

Revised manuscripts should be submitted within three months of a request for revision; they will otherwise be treated as new submissions. Please contact us if a 3-months time frame is not sufficient for the revisions so that we can discuss the revisions further.

2) individual production quality figure files as .eps, .tif, .jpg (one file per figure).

Please download our Figure Preparation Guidelines (figure preparation pdf) from our Author Guidelines pages
<https://www.embopress.org/page/journal/14693178/authorguide> for more info on how to prepare your figures.

4) a complete author checklist, which you can download from our author guidelines (<<https://www.embopress.org/page/journal/14693178/authorguide>>). Please insert information in the checklist that is also reflected in the manuscript. The completed author checklist will also be part of the RPF.

5) Please note that all corresponding authors are required to supply an ORCID ID for their name upon submission of a revised manuscript (<<https://orcid.org/>>). Please find instructions on how to link your ORCID ID to your account in our manuscript tracking system in our Author guidelines (<<https://www.embopress.org/page/journal/14693178/authorguide#authorshipguidelines>>)

6) The Supplementary information is in the correct format (Figure EVx, Dataset EVx, Appendix) but I suggest to add a title to the first page of the Appendix.

7) Before submitting your revision, primary datasets produced in this study need to be deposited in an appropriate public database (see <<https://www.embopress.org/page/journal/14693178/authorguide#datadeposition>>).

Specifically, we would kindly ask you to provide public access to the following datasets:

- RNA seq analysis
- Glyco- and Phosphoproteomics

The accession numbers and database should be listed in a formal "Data Availability " section (placed after Materials & Method) that follows the model below (see also <<https://www.embopress.org/page/journal/14693178/authorguide#datadeposition>>). Please note that the Data Availability Section is restricted to new primary data that are part of this study.

Data availability

8) Our journal encourages inclusion of *data citations in the reference list* to directly cite datasets that were re-used and obtained from public databases. Data citations in the article text are distinct from normal bibliographical citations and should directly link to the database records from which the data can be accessed. In the main text, data citations are formatted as follows: "Data ref: Smith et al, 2001" or "Data ref: NCBI Sequence Read Archive PRJNA342805, 2017". In the Reference list, data citations must be labeled with "[DATASET]". A data reference must provide the database name, accession number/identifiers and a resolvable link to the landing page from which the data can be accessed at the end of the reference. Further instructions are available at <<https://www.embopress.org/page/journal/14693178/authorguide#referencesformat>>.

9) Regarding data quantification:

- Please ensure to specify the name of the statistical test used to generate error bars and P values, the number (n) of independent experiments underlying each data point (not replicate measures of one sample), and the test used to calculate p-values in each figure legend. Discussion of statistical methodology can be reported in the materials and methods section, but figure legends should contain a basic description of n, P and the test applied.

IMPORTANT: Please note that error bars and statistical comparisons may only be applied to data obtained from at least three independent biological replicates. If the data rely on a smaller number of replicates, scatter blots showing individual data points are recommended.

- Graphs must include a description of the bars and the error bars (s.d., s.e.m.).

10) As part of the EMBO publication's Transparent Editorial Process, EMBO reports publishes online a Review Process File to accompany accepted manuscripts. This File will be published in conjunction with your paper and will include the referee reports, your point-by-point response and all pertinent correspondence relating to the manuscript.

I look forward to seeing a revised version of your manuscript when it is ready. Please let me know if you have questions or comments regarding the revision.

REFeree REPORTS

Referee #1:

In this manuscript the authors use the spontaneously immortalized human epidermal keratinocyte, HaCaT, to study the role of GalNAc-T's in epithelial homeostasis and differentiation. They show that GalNAc-T1, T2, and T3 are the three predominantly expressed GalNAc-T's expressed in HaCaT and primary keratinocytes. They subsequently generate knockouts of each using both ZFN nuclease and Crispr-Cas9 (GalNAc-T3 only) methods. The KO cells were induced to form a stratified squamous epithelium in culture, and each showed distinct phenotypes. To better understand the underlying pathways affected by deletion of the individual GalNAc-T's, the authors performed global transcriptomic, differential glycoproteomic, and differential phosphoproteomic analyses on the KO cells. The glycoproteomic analysis revealed the enzymes modified distinct substrates with a small overlap, and the transcriptomic and phosphoproteomic analyses revealed that different underlying pathways were being affected. Gene ontology analysis suggested that the GalNAc-T1 KO affected mainly members of the endomembrane system, GalNAc-T2 KO affected mainly cell-ECM adhesion, while GalNAc-T3 KO affected pathways involved in epithelial differentiation. Significantly, the authors validated the effect of loss of a specific GalNAc-T on the function of specific target proteins.

These are well done studies that confirm the distinct function of three GalNAc-T's in a variety of biological functions which will be of interest to a wide variety of researchers from multiple fields. A few concerns need to be addressed:

1. Figure EV3: The lysotracker staining in the T1 KO cells is very hard to see, so it is difficult to conclude from this data that there is less co-localization of pHrodo-labeled LDL and lysosomes. Is it possible that there is a defect in lysosomal pH in the T1 KO cells?
2. Figures 4L versus EV5: Why does T3 KO result in increased NICD staining in Figure 4L, but

decreased NICD staining in Figure EV5?

3. Top of page 14: The authors suggest that there is significant overlap between the glycosites identified by the two methods (dimethyl labeling and TMT) and they refer to "Fig S6", which this reviewer could not find. In contrast, Dataset EV4 (second tab) shows that only 13 glycopeptides were identified by both methods out of nearly 400 total. The authors suggest that, "This can be partly explained by potential over glycosylation in cells with truncated O-glycans", although if this were the case, more peptides would be identified with the dimethyl labeling (SC samples) than TMT, but the opposite happened with the T1 samples. Unless there is more data suggesting that the two methods are identifying more common sites (i.e. the missing Fig S6), the authors should revise this section to indicate that the methods are different and seeing different things.

Minor comments:

4. Figure 1C: Were none of the other GALNTs detected?

5. Figure 1D: The upper (skin) and lower (HaCaT organ culture) images look quite different, not only the intensity of GalNAc-T staining, but also DAPI staining. Are there better images, or can the authors explain the differences? Also, unlike the description in the text, GalNAc-T2 appears to be basal in the organ culture, but is more widespread in the skin.

6. Dataset EV1: The GALNT3 KO tab shows that levels of GALNT3 are increased. Based on the text, I assume this should be GALNT5? The authors should check for other typos in the datasets.

7. Figure EV1 legend: It would be helpful to define "sc" and "+ST" in the legend.

8. Figure 2B: The Venn diagram needs to be explained in the legend.

9. Dataset EV3: Are there any significant/interesting changes in the proteome that may implicate additional pathways altered in the KO cells?

10. Page 12, last 2 sentences of Results: "Appendix Figure S1" should be "Figure EV5".

11. Page 13, bottom: "With these techniques, we unambiguously identified 67, 88, and 63 single-GalNAc/GalGalNAc glycosites for GalNAc-T1, GalNAc-T2, and GalNAc-T3, respectively." Figure 3D suggests 49, 60, and 50, respectively.

Referee #2:

The manuscript O-glycan initiation directs distinct biological pathways and controls epithelial differentiation provides in-depth analysis of the impact of loss of GalNAc-T1-3 on epithelial cell function and gene expression. It represents a systems biology approach to an important problem. The data are well presented. Some specific deficiencies are noted below:

1. There are no labels on the figures in the pdf making them hard to evaluate.

1. morphological phenotyping would be improved with arrows and annotations. Even a trained pathologist would have trouble identifying the features presented in figure 1 without arrows and enlargements.

2. The enrichment of protein targets in figure 3 are interesting but hard to assess:
Are the protein abundances taken into account?
Are the number of sites/protein considered in this analysis?
What is the enrichment if protein domains are considered? i.e., EGF-repeats etc.

3. The quality of individual micrographs (fluorescence) is poor. The enlarged version help but without proper labelling (see above) it is hard to point out the poor ones.

Since figures are not labelled other problems may be present that I cannot properly decipher.

In summary, this is nice work but without figure legends, hard to assess.

1st Revision - authors' response

6 December 2020

Point-by-point response

Referee #1:

In this manuscript the authors use the spontaneously immortalized human epidermal keratinocyte, HaCaT, to study the role of GalNAc-T's in epithelial homeostasis and differentiation. They show that GalNAc-T1, T2, and T3 are the three predominantly expressed GalNAc-T's expressed in HaCaT and primary keratinocytes. They subsequently generate knockouts of each using both ZFN nuclease and Crispr-Cas9 (GalNAc-T3 only) methods. The KO cells were induced to form a stratified squamous epithelium in culture, and each showed distinct phenotypes. To better understand the underlying pathways affected by deletion of the individual GalNAc-T's, the authors performed global transcriptomic, differential glycoproteomic, and differential phosphoproteomic analyses on the KO cells. The glycoproteomic analysis revealed the enzymes modified distinct substrates with a small overlap, and the transcriptomic and phosphoproteomic analyses revealed that different underlying pathways were being affected. Gene ontology analysis suggested that the GalNAc-T1 KO affected mainly members of the endomembrane system, GalNAc-T2 KO affected mainly cell-ECM adhesion, while GalNAc-T3 KO affected pathways involved in epithelial differentiation. Significantly, the authors validated the effect of loss of a specific GalNAc-T on the function of specific target proteins.

These are well done studies that confirm the distinct function of three GalNAc-T's in a variety of biological functions which will be of interest to a wide variety of researchers from multiple fields. A few concerns need to be addressed:

Query 1. Figure EV3: The lysotracker staining in the T1 KO cells is very hard to see, so it is difficult to conclude from this data that there is less co-localization of pHrodo-labeled LDL and lysosomes. Is it possible that there is a defect in lysosomal pH in the T1 KO cells?

Response 1. We agree that the illustrations could be optimized, as the laser intensity has been set not to oversaturate the brightest images, making LysoTracker poorly visible in GALNT1 KO. LysoTracker does, however, label small vesicles in GALNT1 KO. With that said, we should point out, that we only observed LDL localizing in a proportion of small vesicles in GALNT1 KO making them more difficult to visualize. This is in contrast to the larger vesicles observed in wild type cells.

We acknowledge that the lysosomal pH could be affected. We would, however, not expect that the decrease in intracellular pH observed in GALNT1 KO cells affects the lysosomal pH and hence the detection of LysoTracker.

Action 1. We have now increased the brightness of the red channel in Figure EV3 to enhance visibility.

Query 2. Figures 4L versus EV5: Why does T3 KO result in increased NICD staining in Figure 4L, but decreased NICD staining in Figure EV5?

Response 2. We apologize that we failed to describe these differences clearly. The text is now updated to include an improved description and discussion of the differences between Figure 4L and EV5.

There are a few major differences between the two experiments. Firstly, different keratinocyte cell lines are used to monitor NICD cleavage. In Fig 4L, non-labeled HaCaT keratinocytes are co-cultured with immortalized primary keratinocytes (N/TERT1) labeled with cell-tracker (red). As seen in these cultures, HaCaT GALNT3 KO cells do not display increased Notch1 cleavage (increase in NICD), when compared to HaCaT WT cells. In fact, the proportion of HaCaT GALNT3

KO cells undergoing Notch1 activation is somewhat lower compared to HaCaT WT (see the updated Fig 4L with boundaries between two cell types marked). This is consistent with the decreased NICD cleavage observed when HaCaT cells are grown in 3D in Fig EV5. In Fig 4L we, however, observed increased Notch1 cleavage in the N/TERT1 cells when co-cultured with HaCaT GALNT3 KO cells. This co-culture experiment is designed to present the Notch ligand on HaCaT cells and induce trans activation of Notch in N/TERT1 cells. The results suggest that absence of GalNAc-T3 in the ligand presenting cells (HaCaT) causes premature activation of Notch1 in the receiving cell (N/TERT1 cells).

Fig EV5 shows 3D differentiated epidermis created with HaCaT cells alone. In this situation, we again observe decreased Notch1 cleavage in GalNAc-T3 KO cells, but we do not see any increased activation of Notch1 as observed in the N/TERT1 cells. As mentioned in the text (page 17, line 8), we do not have a clear explanation for this, but speculate that it could be due to a GalNAc-T3 mediated effect on the balance between cis and trans-activation of Notch1. In this context, it is important to remember that human keratinocytes express multiple Notch isoforms and several different ligands, and it is therefore difficult with the current setup to assess the mechanistic details of how GalNAc-T3 affects differentiation.

Action 2. Text is now updated to include a description of the differences between Figure 4L and EV5. In addition, we marked the boundaries between HaCaT and N/TERT-1 cells in the cleaved Notch1 staining images in Fig 4L, to better see the level of Notch1 activation in the two cell types. The text on page 12 was updated to the following:

“In order to address the potency of Jagged-1^{T3KO} to induce Notch1 signalling, we co-cultured HaCaT WT or GALNT3 KO keratinocytes with N/TERT1 immortalized keratinocytes, and assessed Notch1 activation by nuclear expression of cleaved Notch1 intracellular domain (NICD). In addition, we evaluated differentiation by cytokeratin 10 staining. Co-culture of N/TERT1 WT and HaCaT GALNT3 KO cells induced robust differentiation of N/TERT1 cells, which were mostly organized into compact piles, strongly positive for nuclear cleaved NICD, and increased keratin 10 expression (Fig 4L-N). In contrast, co-culture of WT HaCaT and N/TERT1 keratinocytes only induced limited keratin 10 expression and failed to induce organized piles of keratinocytes positive for NICD (Fig 4L-N). Furthermore, a lower level of Notch1 intracellular domain cleavage was observed in HaCaT WT, with fewer cells positive for cleaved NICD in HaCaT GALNT3 KO cells. The combined results support the potential role of GalNAc-T3-specific O-glycosylation on differentiation possibly through fine-tuning Jagged-Notch interactions. We also investigated Notch1 signaling in the organotypic models by visualizing localization of cleaved NICD, which was primarily detectable in the nuclei of suprabasal cell layers of WT HaCaT skin (Fig EV5). Compared to WT, little Notch1 signaling was observed in suprabasal layers of GALNT3 KO organotypic models (Fig EV5). This is consistent with the delayed differentiation phenotype, and likely signifies the contribution of multiple GalNAc-T3 targets to tissue development.”

Query 3. Top of page 14: The authors suggest that there is significant overlap between the glycosites identified by the two methods (dimethyl labeling and TMT) and they refer to "Fig S6", which this reviewer could not find. In contrast, Dataset EV4 (second tab) shows that only 13 glycopeptides were identified by both methods out of nearly 400 total. The authors suggest that, "This can be partly explained by potential over glycosylation in cells with truncated O-glycans", although if this were the case, more peptides would be identified with the dimethyl labeling (SC samples) than TMT, but the opposite happened with the T1 samples. Unless there is more data suggesting that the two methods are identifying more common sites (i.e. the missing Fig S6), the authors should revise this section to indicate that the methods are different and seeing different things.

Response 3. We thank the reviewer for the comment and apologize for the wrong reference – it should refer to Appendix Figure S1. We do refer to overlap between protein substrates, because the overlap at individual glycosite level is indeed very small. In fact, GalNAc-T1 is the only case where more substrates were identified using the TMT method. It is the opposite for GalNAc-T2 and -T3 where we also see around 30 % of protein substrate overlap. As the reviewer points out, it is important to acknowledge the methodological differences, including the chemical properties of the two labels differently affecting the resulting sample complexity as well as the dynamic range of detected glycopeptides, as well as the different lectin enrichment approaches.

Action 3. The text on page 14 was corrected to the following:

“We found limited overlap between the identified glycosites between each GalNAc-T isoform, supporting the validity of the strategy and identified candidates. Comparison of glycoprotein and glycosite identities for each isoform between the two methods has shown variable overlap (Appendix Fig S1, Dataset EV4). Almost a third of glycoprotein substrates for GalNAc-T2 and GalNAc-T3 identified with TMT labeling were also identified with dimethyl labeling, whereas very little overlap was observed for GalNAc-T1 (Appendix Fig S1). However, the majority of commonly identified protein substrates were glycosylated at different positions using the two methods. This is not surprising given the different genetic backgrounds of the cells used for the analysis, as well as the distinct methodological approaches affecting the chemical properties and the dynamic range of detected glycopeptides. Transcriptomic and proteomic differences from WT cells should also be taken into account.”

Minor comments:

Query 4. Figure 1C: Were none of the other GALNTs detected?

Response 4. Yes, that is correct – RPKM expression values for GALNT4, GALNT8, GALNT9, GALNT13, GALNT15, GALNT16, and GALNT19 were well below 1 RPKM. The individual RPKM values for the different GALNT genes can be found in Dataset EV1.

Query 5. Figure 1D: The upper (skin) and lower (HaCaT organ culture) images look quite different, not only the intensity of GalNAc-T staining, but also DAPI staining. Are there better images, or can the authors explain the differences? Also, unlike the description in the text, GalNAc-T2 appears to be basal in the organ culture, but is more widespread in the skin.

Response 5.

The figure has now been improved by including more directly comparable images. The differences in the submitted figure are mainly due to different imaging techniques, with skin imaged using widefield microscopy and organotypic models using confocal microscopy, as well as different magnifications. Regarding the DAPI stain, we occasionally see poor dye penetrance in frozen tissue sections, which can be improved with sucrose treatment prior to freezing.

GalNAc-T2 staining in fact seems a bit more widespread in the organ culture, although primarily located in the basal layer, where most intense staining is observed (lower panel), and more selective in skin (upper panel).

Action 5. The images of GalNAc-T stainings in skin previously obtained with a widefield fluorescent microscope have been replaced with confocal micrographs in Figure 1.

The text on page 4 has been corrected to the following:

“Immunocytochemistry showed the localization of GalNAc-T1, -T2, and -T3; human skin and HaCaT 3D models expressed GalNAc-Ts in a similar expression pattern, with GalNAc-T2 primarily expressed in basal cells and broader expression of GalNAc-T1 and -T3 in all epithelial layers (Fig 1D).”

Query 6. Dataset EV1: The GALNT3 KO tab shows that levels of GALNT3 are increased. Based on the text, I assume this should be GALNT5? The authors should check for other typos in the datasets.

Response 6. The GALNT3 tab in fact shows decreased expression of GALNT3 (log(fold change) = -1.629412048). That is because the knock out procedure truncates and functionally inactivates genes of interest, however, truncated mRNA is still expressed and not completely gone. Regarding GALNT5, it does not show up as statistically significant in the analysis when all three GALNT3 KO clones are considered, therefore is not present in the respective tab. The mention in the text is based on pulling out all GALNT genes from the dataset without doing statistical analysis.

Action 6. We have included the graph reflecting increase in GALNT5 expression in the appendix. In addition, we have updated the figure legend to better describe the statistical analysis.

“Dataset EV1. RNA Seq. Gene expression data in RPKM is presented. In the “all genes normalized” tab, missing values were replaced with 0.001, and data analyzed using the CLC Genomics Workbench (Qiagen). In the individual KO tabs, statistically significant gene expression changes considering all three knock out clones of individual GALNTs (R package EdgeR analysis) are shown (filtered for at least 1 RPKM in at least one of the compared entries). $\log_2(\text{KO}/\text{WT})$.”

Appendix Figure S2. Expression of GALNT genes (> 1 RPKM) in GALNT KO cell lines.

Query 7. Figure EV1 legend: It would be helpful to define "sc" and "+ST" in the legend.

Action 7. The figure legend was corrected to the following:

Figure EV1. Characterization of GALNT KO cell lines. GALNT KO cell lines in WT or COSMC KO (sc, “Simple Cell”) background were stained for GalNAc-T1, GalNAc-T2, GalNAc-T3, as well as T (with (ST (sialyl-T) + T) or without (T) neuraminidase treatment), Tn, and STn (sialyl-Tn) glycoforms using monoclonal antibodies and lectins. Scale bar – 10 μm .

Query 8. Figure 2B: The Venn diagram needs to be explained in the legend.

Action 8. The figure legend was corrected to the following:

Figure 2. Phosphoproteomic analysis. (A) Strategy for differential phosphoproteomics of HaCaT GALNT KO cell lines, where the Gaussian distributions and the Venn diagram represent selection criteria. (B) Analysis of differentially phosphorylated peptides. Upregulated or downregulated phosphopeptides in GALNT KO were considered significant, if phosphopeptides were identified in both clones more than 2x SD away from the median, and given the same phosphopeptides were within the normal variation of two wild type biological replicates. The Venn diagram illustrates the selection criteria, where the color scheme corresponds to the Gaussian distribution illustration above. The bar graph depicts numbers of unique differential phosphosites in the individual GALNT KO. (C) The Venn diagram represents overlap between differential phosphosites identified in the different GALNT isoform KO. (D) GO terms (DAVID) associated with phosphoproteins identified in the 3 differential datasets. p value (Benjamini multiple comparison adjusted) of 0.05 is marked with a dashed line.

Query 9. Dataset EV3: Are there any significant/interesting changes in the proteome that may implicate additional pathways altered in the KO cells?

Response 9. We do see very limited proteome changes, if we stick with the identical analysis strategy used for the phosphoproteomic dataset. However, if we choose less strict criteria for the analysis, we see additional associations such as cellular response to external stimuli. For example, we find upregulation of proteins involved in wound healing and platelet activation in GALNT1 KO,

whereas type I interferon/antiviral response and antigen presentation were downregulated both in GALNT2 KO and GALNT3 KO. In addition, we observe some feedback responses to pathways affected by the individual GalNAc-Ts. An illustrative example is the upregulation of proteins involved in cell to matrix adhesion in GALNT2 KO that display compromised cell-matrix interactions. However, we have refrained from such assumptions in the manuscript, given the lack of statistical power and further experimental investigation.

Query 10. Page 12, last 2 sentences of Results: "Appendix Figure S1" should be "Figure EV5".

Action 10. Corrected, thank you.

Query 11. Page 13, bottom: "With these techniques, we unambiguously identified 67, 88, and 63 single- GalNAc/GalGalNAc glycosites for GalNAc-T1, GalNAc-T2, and GalNAc-T3, respectively." Figure 3D suggests 49, 60, and 50, respectively.

Response 11. The numbers mentioned in the text are correct, as are the ones in the figure. The latter numbers reflect the targets exclusively found in individual isoforms, as suggested in the Venn diagram. If the targets overlapping with those found in other isoforms are included, the numbers add up to those referred in the text.

Referee #2:

The manuscript O-glycan initiation directs distinct biological pathways and controls epithelial differentiation provides in-depth analysis of the impact of loss of GalNAc-T1-3 on epithelial cell function and gene expression. It represents a systems biology approach to an important problem. The data are well presented. Some specific deficiencies are noted below:

Query 1. There are no labels on the figures in the pdf making them hard to evaluate.

Response 1. We apologize for the inconvenience. It was wrong of us to presume the figures would be labelled in the automatic generation of the PDF.

Query 2. Morphological phenotyping would be improved with arrows and annotations. Even a trained pathologist would have trouble identifying the features presented in figure 1 without arrows and enlargements.

Response 2. We have included annotations in Figure 1E with relevant links in the legend and the main text.

Action 2. The text on page 5 was updated to include links to marked elements in Figure 1E, and the figure legend was updated to the following:

“(E) Phenotypic characterization of organotypic models made with HaCaT WT or GALNT1/2/3 KO keratinocytes. IHC of tissue sections stained for differentiation marker keratin 10 (upper panel) or proliferation marker Ki67 (lower panel). Scale bar – 50 μ m. Red arrows – flattened cells; red asterisks – K10-negative region in suprabasal/granular layers; purple asterisks – pyknotic nuclei; green asterisks – increase in Ki67 positive cells.”

Query 3. The enrichment of protein targets in figure 3 are interesting but hard to assess:

Are the protein abundances taken into account?

Are the number of sites/protein considered in this analysis?

What is the enrichment if protein domains are considered? i.e., EGF-repeats etc.

Response 3. Figure 3 summarizes results from lectin enrichment experiments, where only glycosylated peptides are captured, and thus protein abundances are not considered, however, peptide data prior to Jacalin enrichment is available for TMT-labelled samples, which shows mostly unchanged protein levels of top targets presented in Table 1.

We have in previous exploratory O-glycoproteomic studies (Steentoft et al., EMBO J 2013, King et al., Blood Adv 2017) looked at number and distribution of glycosites on proteins, and found that the largest proportion of proteins are modified with one or several glycosites. Compared to broad mapping studies, with characterization of hundreds of glycosites, differential glycoproteomic datasets contain much lower numbers of reported targets. Therefore, we did not find such analysis relevant in the current study.

As discussed in the text, proteins containing fibronectin type 3 domains were more likely to be glycosylated by GalNAc-T2, whereas proteins containing EGF-like domains were over-represented in GalNAc-T1 and GalNAc-T3 targets. The associations are based on specific targets containing such domains. However, the location of glycosite in respect to the domain was not consistent. Of the six GalNAc-T2 targets containing FN3 domains, five targets contained O-glycosites between or next to the first/last FN3 domain, whereas one glycosite was located within the FN3 domain. In contrast, GalNAc-T1 and GalNAc-T3 glycosites were located far away from the EGF-like domains. In general, it should be mentioned that the proportion of targets is too low to make a definitive conclusion.

Action 3. We have analyzed the proteome of the TMT-labelled sample prior to Jacalin lectin enrichment, and included the average protein quantification in the 3 knock out clones compared to wild type for top targets presented in Table 1, as well as protein coverage by peptides. We excluded targets with protein ratios below 0.75.

Query 4. The quality of individual micrographs (fluorescence) is poor. The enlarged version help but without proper labelling (see above) it is hard to point out the poor ones.

Response 4. We agree that certain micrograph image elements could benefit from higher resolution, however, it is challenging to maintain the original quality given the file size requirements for initial submission.

Since figures are not labelled other problems may be present that I cannot properly decipher.

In summary, this is nice work but without figure legends, hard to assess.

2nd Editorial Decision

14 February 2020

Thank you for the submission of your revised manuscript to EMBO reports. I apologize for the unusual delay in handling your manuscript. We have now received the report from former referee #1 who supports publication of the manuscript in EMBO reports and I have editorially checked your response to the concerns from referee 2, who was not available anymore.

Given the support from referee 1 on the revised version and the fact that referee 2 has supported publication of the first version after minor revision, I thus decided to move forward with publication.

Browsing through the manuscript myself, I noticed a few editorial things that we need before we can proceed with the official acceptance of your study.

- 1) Your manuscript will be published as Report. Therefore, please combine the Results and Discussion sections. This will also help to shorten the manuscript text to get closer to our character limit of 25,000 plus/minus 2,000 characters (excl. materials and methods and references) by eliminating some redundancy that is inevitable when discussing the same experiments twice.
- 2) Fig 4I+N: please provide scale bars for the magnification boxes.
- 3) I attach to this email a related manuscript file with comments by our data editors. Please address all comments and upload a revised file with tracked changes with your final manuscript submission. I have also taken the liberty to make some changes to the Abstract. Could you please review it?

4) Finally, EMBO reports papers are accompanied online by A) a short (1-2 sentences) summary of the findings and their significance, B) 2-3 bullet points highlighting key results and C) a synopsis image that is 550x200-400 pixels large (width x height). You can either show a model or key data in the synopsis image. Please note that the size is rather small and that text needs to be readable at the final size. Please send us this information along with the revised manuscript.

REFEREE REPORT

Referee #1:

The authors have addressed all of the concerns I raised in the initial review. I have no further concerns.

2nd Revision - authors' response

3 March 2020

The authors performed all minor editorial changes.

Corresponding Author Name: Hans H. Wandall

Manuscript Number: EMBOR-2019-48885V2